# Internal Validation of MaSTR™ Probabilistic Genotyping Software for the Interpretation of 2–5 Person Mixed DNA Profiles

**DOI:** 10.3390/genes13081429

**Published:** 2022-08-11

**Authors:** Michael S. Adamowicz, Taylor N. Rambo, Jennifer L. Clarke

**Affiliations:** College of Agricultural Sciences & Natural Resources, University of Nebraska-Lincoln, 103 Agriculture Hall, Lincoln, NE 68583-0702, USA

**Keywords:** probabilistic genotyping, DNA mixtures, STR analysis, likelihood ratios, MaSTR™, validation

## Abstract

Mixed human deoxyribonucleic acid (DNA) samples present one of the most challenging pieces of evidence that a forensic analyst can encounter. When multiple contributors, stochastic amplification, and allele drop-out further complicate the mixture profile, interpretation by hand becomes unreliable and statistical analysis problematic. Probabilistic genotyping software has provided a tool to address complex mixture interpretation and provide likelihood ratios for defined sets of propositions. The MaSTR™ software is a fully continuous probabilistic system that considers a wide range of STR profile data to provide likelihood ratios on DNA mixtures. Mixtures with two to five contributors and a range of component ratios and allele peak heights were created to test the validity of MaSTR™ with data similar to real casework. Over 280 different mixed DNA profiles were used to perform more than 2600 analyses using different sets of propositions and numbers of contributors. The results of the analyses demonstrated that MaSTR™ provided accurate and precise statistical data on DNA mixtures with up to five contributors, including minor contributors with stochastic amplification effects. Tests for both Type I and Type II errors were performed. The findings in this study support that MaSTR™ is a robust tool that meets the current standards for probabilistic genotyping.

## 1. Introduction

Probabilistic genotyping using dedicated software to interpret mixed deoxyribonucleic acid (DNA) profiles and provide likelihood ratio (LR) data addressing the probability of a match between the evidence and person(s) of interest (POI) has become an accepted and ever more common tool in forensic science laboratories [1]. The use of LRs based on the evaluation of evidence given two or more mutually exclusive sets of propositions [2,3] instead of binary approaches, such as the combined probability of inclusion/combined probability of exclusion (CPI/CPE) [4,5], has been advocated for some time [6,7]. However, LR calculations can be challenging to set up, particularly with complex mixture samples that include three or more individual contributors as well as degraded, inhibited, or low template DNA sources [8]. A variety of commercial and open-source probabilistic genotyping software is now available to forensic DNA laboratories to choose from [1]. These software applications fall into two categories: semi-continuous and fully continuous. They differ in the data that each type uses in their respective algorithms. Semi-continuous systems use the alleles that are detected in a mixture profile, combining them in all possible combinations and including probabilities of allele drop-in and drop-out. Fully continuous systems use allele data and additional information such as stutter percentages and electropherogram peak heights. Regardless of which is used, both types of probabilistic systems result in LR calculations that inherently utilize more information than the previous binary approaches [8].

To properly utilize probabilistic genotyping software, it must be thoroughly validated, like any other part of the forensic analysis process. There are a variety of published internal validation guidelines and standards for probabilistic genotyping systems, including those put forth by the Scientific Working Group on DNA Analysis Methods (SWGDAM) [9], the DNA Commission of the International Society for Forensic Genetics [10], and the Approved American National Standard/AAFS Standards Board (ANSI/ASB) [11]. These sources stipulate that internal validation studies need to include and address a series of common experimental features. Accuracy, sensitivity, specificity, precision, software input parameters, allele sharing, locus and allele drop-out, stutter, peak height variation, degraded template DNA, and evaluation of multiple propositions should all be included. Thorough validation should also pay particular attention to finding the limits of the software, at which point the results are no longer reliable or reproduceable. Validation studies have been published on a variety of semi- and fully continuous probabilistic genotyping software programs. Some of these include the semi-continuous Lab Retriever [12] and LRMix/LRMixStudio [13,14,15] and the fully continuous TrueAllele^®^ [16,17,18,19], EuroForMix [20,21], and STRmix™ [22,23,24].

The MaSTR™ software uses a fully continuous probabilistic genotyping approach which incorporates a Markov Chain Monte Carlo (MCMC) method with the Metropolis-Hastings algorithm [25,26,27]. The application and implementation of MCMC to obtain weighted genotypes from text files that contain allele calls and peak height information has been well documented in the scientific literature [22,28,29]. The text files are obtained from separate genotyping software such as GeneMarker^®^HID, GeneMapper^®^ID-X, or OSIRIS after evaluation by a forensic analyst for size and allele calling, peak adjudication, and estimation of the number of contributors [30]. In addition to peak height, MaSTR™ models incorporate stutter, allelic drop-in and drop-out, and Bayesian prior assumptions: allele probabilities and statistical formulae used in the calculation and algorithms. Likelihood ratios are calculated using weighted genotypes obtained from the MCMC process. Additional functions include (1) an automated comparison of all genotypes in the elimination database to alert the laboratory if any of the genotypes in the staff elimination database have a positive likelihood ratio for inclusion in the mixture; (2) an option to calculate the LR of the person of interest versus a potential relative in addition to the LR of the person of interest versus a random individual from the population; (3) the option to analyze replicates of a mixture; and (4) results for each analysis including the LR for the person(s) of interest, text file export of weighted genotypes providing output for most-likely unknown contributor profiles for entry in Combined DNA Index System (CODIS), LR plots of 100 random in silico genotypes for each population (400–900 genotypes, depending on the population data selected), mixture ratio plots, which will provide a visual of ratio per chain and indicate if one or more chains were not in convergence, mixture ratio histograms, degradation trace, and histogram plots. The study described in this paper was designed to address the internal validation requirements of a probabilistic genotyping system and find the limits of the MaSTR™ software using a wide range of mixed DNA combinations of known composition, ratios, and electropherogram peak heights, including stochastic amplification of contributor DNA similar to those found in casework evidentiary samples. Mixtures of individuals with both high and low levels of allele sharing were created and a variety of LR propositions were examined using both known contributors and non-contributors to test for Type I (LR < 1 for a true contributor) and Type II (LR > 1 for a true non-contributor) errors [9].

## 2. Materials and Methods

### 2.1. DNA Procurement and Preparation

All human contributor DNA used in this study was purchased as de-identified extracts from the Nebraska BioBank (Omaha, NE, USA). No information regarding the race, ethnicity, or genetic ancestry of the donors was included. DNA was extracted at the Nebraska BioBank from the blood of 40 individuals using the Qiagen (Valencia, CA, USA) QIAcube and the QIAamp^®^ 96 DNA QIAcube HT kit optimized for downstream genomic assays. Addition of proK and AL buffer and incubation at 56 °C resulted in lysis of the cells. The extracts were added to the spin columns in the Qiacube for washing and elution of DNA. The DNA was washed via a salt wash (AW1) followed by a salt-eliminating wash (AW2), then an ethanol wash. DNA was eluted from the column via a low ionic elution buffer. The resulting DNA was quantified using the NanoDrop™ One (ThermoFisher, Santa Clara, CA, USA) micro-volume UV-Vis spectrophotometer at the Nebraska BioBank (Omaha, NE, USA). To further determine the concentration of DNA in each extract after they were delivered, the extracts were quantified using a 7500 real-time polymerase chain reaction (rtPCR) system (Applied Biosystems) and the Quantifiler™ Human DNA Quantification Kit (Applied Biosystems, Foster City, CA, USA) according to the manufacturer’s recommendations [31].

### 2.2. DNA Amplification

All DNA samples were amplified for short tandem repeat (STR) analysis using the Promega (Madison, WI, USA) PowerPlex^®^ Fusion 5C kit following the manufacturer’s protocols [32]. The target initial quantity of DNA for amplification was approximately 500 pg per contributor and was subsequently adjusted according to the requirements of the specific experimental conditions, i.e., specific mixture ratios. The amplification reaction final volume was 25 µL and thermal cycling was performed in a GeneAmp PCR System 9700 (Applied Biosystems, Foster City, CA, USA) using maximum ramp speed for 30 cycles. All amplification sets included a positive control (cell line 2800M DNA, included in kit) and a negative control (deionized water used for sample dilutions).

### 2.3. Capillary Electrophoresis and STR Genotyping

Separation and detection of the PCR products was performed using a 3130-Avant Genetic Analyzer (Applied Biosystems, Foster City, CA, USA) with Foundation Data Collection software v4.0, following all manufacture recommended protocols [33]. All DNA was prepared for injection in a loading cocktail of Hi-Di^TM^ formamide (Applied Biosystems, Foster City, CA, USA) (9.5 µL/sample) and WEN internal lane standard 500 (Promega, Madison, WI, USA) (0.5 µL/sample). All injections were performed at 3 kV for 5 s. Separation occurred in POP4 Polymer (Applied Biosystems, Foster City, CA, USA). Analysis of all the injection products was performed using SoftGenetics^®^ LLC (State College, PA, USA) GeneMarker^®^ HID v2.9.0 software. The analytical threshold for the instrument used was previously determined to be a minimum of 30 relative fluorescence units (RFUs). All the two-person mixture samples were analyzed with a higher analytical threshold of 50 RFUs, as were the three-person mixtures, except for those that were maximally diluted 1:8 (~6–63 pg/contributor), which used an analytical threshold of 30 RFUs. All the four- and five-person mixtures also used an analytical threshold of 30 RFUs. The differing analytical thresholds reflected the prevalence of low-level peaks in each condition and the desire to include allele dropout in some of the two- and three-person combinations which was not seen when those mixture profiles were analyzed at 30 RFUs. This approach further challenged the MaSTR™ software at each mixture ratio.

### 2.4. DNA Mixture Preparation

Mixtures were created using two, three, four, or five of the single source contributors. Two different strategies were used in creating the two- and three-person mixtures. After each single source donor was genotyped, an appropriate number of mixture contributors (two or three) were selected that showed minimal or maximal allele sharing among the available genotypes, to test the MaSTR™ software under these different conditions. Two different two-person mixture combinations were created, one in which the contributors shared a total of five alleles among five loci (low share) and one in which the contributors shared 19 alleles among 18 loci (high share). The same process was used for the three-person mixtures, resulting in one combination with 16 alleles shared among 15 loci between the three contributors (low share) and a different combination of contributors showing 31 alleles shared among 21 loci (high share). The four- and five- person mixtures were created by randomly selecting contributors and combining them without regard to allele sharing, providing random combinations of alleles. Only one single source contributor, 1683, was used twice (Table 1).

This gave a total of 18 different single source genotypes used to create six separate mixture genotypes. Two-person mixtures were made in the following ratios: 1:1, 1:2, 1:3, 1:5, and 1:10 (high- and low-share each). Three-person mixtures were made in the following ratios: 1:1:1, 1:1:2, 1:3:5, and 1:2:10 (high- and low-share each). Four-person mixtures were made in the following ratios: 1:1:1:1, 1:2:2:5, and 1:1:3:10. Five-person mixtures were made in the following ratios: 1:1:1:1:1, 1:2:2:5:10, and 1:1:5:5:10 (Table 2).

When creating the mixtures, the largest undiluted single contributor quantity of DNA template was always set at approximately 500 pg. The amount of DNA added to the two person mixtures directly reflected the ratio between the two; however, for the three-, four-, and five-person mixtures the builds were slightly different. The highest quantity undiluted contributor was held at approximately 500 pg and the lowest undiluted contributor was approximately 50–100 pg, depending on the relevant ratio. All other contributor quantities were in relation to the lowest contributor(s) (Table 2).

The mixture ratio balance for each contributor was assessed by multiple methods prior to final amplifications. As previously described, all peak heights were assessed using SoftGenetics^®^ LLC (State College, PA, USA) GeneMarker^®^ HID v2.9.0 software. The peak heights of each PCR amplified single-source genotype selected for combination were compared to the others in the planned mixture. The amount of input DNA for each contributor was then adjusted so that the peak heights in the mixture should correspond to the desired ratio. Test mixture samples were then assembled, amplified, and capillary electrophoresis was performed in triplicate to assess if the amount of DNA in the respective amplifications gave acceptable peak height ratios or needed further adjustment. Once the test mixture contributors demonstrated peak heights that fit expectations for the desired mixture ratio, those quantities of DNA were used to create the experimental samples. Each final mixture sample was also serially diluted 1:2, 1:4, and 1:8 to examine the same contributor ratios with different overall peak heights, as well as with and without stochastic amplification effects. This resulted in 96 different mixture sample conditions to analyze (Table 2). Every mixture/dilution sample combination was amplified in triplicate to assess the impact of the PCR process induced peak height variances on the MaSTR™ software’s ability to return consistent LR values for a particular combinatorial condition. This provided a large number of mixture samples for each condition for analysis (Table 2). Single source genotypes that were not chosen to combine into the mixtures were used as known non-contributors to challenge the software’s ability to properly identify H_2_ true (LR < 1) values, in addition to the synthetic in silico genotypes generated by MaSTR™.

### 2.5. Graphical User Interface

The MaSTR™ software used a menu-driven graphical user interface (GUI) to set analysis conditions, create savable models and STR panels, examine population frequencies, and calculate the peak height variance factor (including stutter peaks). Uniquely named mixture analysis instructions were created and saved using a menu option termed “Models”, which contained the number of MCMC iterations, length of burn-in and Thin N values, as well as number of chains in the calculation. Additionally, the “Model” creation menu was where options for Drop In coefficient, preprocessing steps, the application of Metropolis or Slice calculations, and whether to use linear, exponential, or no adjustments for DNA degradation were set. A second set of drop-down menus selected the specific STR panels for the amplification kit that was used, including information on each locus in the panel. The analytical threshold value used to determine contributor genotypes was defined in this menu. The population frequency database options were also selectable, with options for using the Budowle et al., 2001 [34], Hill et al., 2013 [35], or Moretti et al., 2016 [36] allele frequencies; however, other databases could be imported. Information on sample analysis processing, such as the number of PCR cycles used in amplification, type of capillary electrophoresis instrument used, the electrophoretic run voltage, and injection time was entered to create a savable Protocol Set. Analysis runs used a separate menu with drop-down options for selecting the created Model and Protocol Set, choice of calculation with or without co-ancestry (Hardy–Weinberg equation, National Research Council II recommendations 4.1 or 4.2) [37,38] and adjustment value for θ. This menu also included selecting a unique identifier for the analysis run, number of contributors, whether to consider relatives in the calculations or not, and entry for the signal (unknown mixture sample), known, reference, and alternate files. It should be noted that in the version of MaSTR™ used, the term “reference” meant the first unknown genotype (POI) to be compared to the mixture sample and “alternates” were sequential additional unknowns.

### 2.6. Protocol Dataset

Prior to the analysis of mixed samples, a protocol dataset was created for the MaSTR™ software to use in the assessment of backwards (N-1) stutter peaks and the calculation of a variance factor. The variance factor is calculated to assess the level and distribution of allele peak height variability, including stutter peaks, of a particular STR amplification kit/capillary electrophoresis instrument combination [39]. Ten single source genotypes were chosen that demonstrated the greatest level of allelic variation available. Each contributor DNA was amplified using an initial quantity of approximately 500 pg with five replicates. The DNA extracts were then diluted 1:2 (250 pg) and 1:4 (125 pg) and amplified again, each with five replicates, yielding a total of 150 analysis DNA extracts with 10 distinct genotypes and a broad range of peak heights and stutter values for the MaSTR software to use. It is of note that no unaccounted allelic peaks greater than or equal to the relevant analytical threshold were detected in the protocol dataset, or in any other sample in this study, indicating zero detectable drop-in peaks.

### 2.7. Likelihood Ratio Calculations

The MaSTR™ software was used to perform all LR calculations using a desktop computer (Intel^®^ Core™ i7-7700 CPU 3.60 GHz 4 core 8 logical processors) with 8 GB RAM operating with Windows 10. This study used versions 1.7–1.11 of the software. Version improvements included faster run times, improved user interface, refinements in data presentation, and additional options in final report formats. All data generated with earlier versions were re-run with v1.11 to confirm concordant results, and all LRs reported herein were calculated with v1.11. It should be noted that the MaSTR™ software calculates both the simple (no specified order of contributors) and overall (specified order of contributors) likelihood ratios; however, only the overall values [28] are reported in this study, except where specifically stated. The simple LRs correspond sub-source level propositions and the overall LRs correspond to sub-sub-source level propositions, as described by Taylor et al. [40]. The version of MaSTR™ used in this study would not report an absolute exclusion (LR = 0), nor did it use a numerical cutoff limit for LRs < 1. MaSTR™ calculated the LR for each unknown contributor, whether entered as a reference or an alternate, and reported that value no matter how low it was. The software settings for analysis used the pre-populated PowerPlex^®^ Fusion panels v1.0, the population database selected was that of Hill et al. [35], the co-ancestry adjustment was that recommended in the NRC II document’s recommendation 4.1 [37] with a θ value of 0.01, unless otherwise specified. The database used contained allele frequencies for African American, Caucasian, Hispanic, and Asian populations, as well as an All Dataset which combines the separate populations [35]. All MaSTR™ calculations in this study are reported using the All Dataset as the population group of each contributor DNA was unknown. MaSTR™ automatically performed calculations for each of the separate populations in the selected database and those results could be accessed if desired. Relatives were not considered. All analyses were performed using eight MCMC chains, but with differing numbers of iterations, depending on the complexity of the mixture [41]. Two-person mixtures were run with both 5000 and 10,000 iterations per chain. Three-person mixtures were run with both 10,000 and 20,000 iterations per chain. Four- and five-person mixtures were run with both 20,000 and 50,000 iterations per chain. The MCMC “burn-in” was set to 20% of the number of iterations per chain, the Metropolis Hastings calculation [25,26] option was used, the software degradation option setting selected was “None”, Thin N was set to 1, the Drop In coefficient was set to 1, and the preprocessing steps option was set to 10,000 for all analysis runs. All MaSTR™ calculations were assessed for precision by performing five replicate analyses. Only the autosomal loci were used by MaSTR™ in this study: amelogenin and DYS391 information was not included.

Likelihood ratios were calculated for several different propositions, depending on the number of contributors to the mixture. These included a POI genotype set as the reference contributor and variable numbers (N-1) of unknown, unrelated contributors set as alternates, which could also include true non-contributors in some analyses as the H_1_ propositions. The H_2_ propositions were set as N unknown individuals (Table 3). Some LR calculations were conditioned on the inclusion of a known contributor in the mixture combinations where the H_1_ and H_2_ propositions were modified to add an assumed known individual to those previously described above (Table 3).

Additionally, a sub-set of three- and four-person mixture samples were selected to test with variable numbers of identified contributors. In each of these tests, LRs were calculated for the true number of contributors plus and minus one to assess the software’s ability to provide data when the wrong, but possible, number of contributors was selected.

The *MaSTR™* software makes two sets of LR calculations for each contributor in a mixture sample: a Simple (sub-source) and Overall (sub-sub-source) likelihood ratio. The *SimpleLR* uses the formula:(1)$$SimpleLR=\frac{\prod_{l}^{}\sum_{j}^{}w_{j}^{l}Pr(S_{j}^{l}|H_{1})}{\prod_{l}^{}\sum_{j^{\prime }}^{}w_{j^{\prime }}^{l}Pr(S_{j^{\prime }}^{l}|H_{2})}$$
where $\prod_{l}$ is the product across each locus l, $S_{j}^{l}$ is the jth genotype set in locus l, and $w_{j}^{l}$ is the weight of the jth genotype set in the lth locus from the MCMC results (40). The Simple LR does not consider the position in the mixture of the tested genotype, and therefore, usually returns a higher LR value. The Overall LR uses the formula:(2)$$LR=\frac{\sum_{i=1}^{N!}\prod_{l}^{}\sum_{j}^{}w_{j}^{l}Pr(S_{j}^{l}|C_{i}H_{1})}{\sum_{i=1}^{N!}\prod_{l}^{}\sum_{j^{\prime }}^{}w_{j^{\prime }}^{l}Pr(S_{j^{\prime }}^{l}|H_{2})}$$
where $C_{i}$ is a specific contributor order tested using N!, N is the number of unknowns, including the tested profile, and $\sum_{i=1}^{N!}$ is the sum over all potential orders of unknown contributors [28,42]. In conditions where related persons are not being tested, the Overall LR can be simplified to:(3)$$LR=\frac{\sum_{i=1}^{N}\prod_{l}^{}\sum_{j}^{}w_{j}^{l}Pr(S_{j}^{l}|D_{i}H_{1})}{N\prod_{l}^{}\sum_{j^{\prime }}^{}w_{j^{\prime }}^{l}Pr(S_{j^{\prime }}^{l}|H_{2})}$$
where $D_{i}$ is the possibility of the POI being the ith contributor to the mixture [42]. All LR values reported in this work are Overall likelihood ratios unless otherwise specified.

### 2.8. Statistical Visualizations

Statistical visualizations of the validation results were done using functions from the libraries ggplot2, dplyr, and tidyverse from the R Statistical Language [43].

## 3. Results

### 3.1. Variance Factor Results

Variance factor values for allele peaks, including backwards stutter peaks, were derived using the protocol dataset for both the 30 RFUs and 50 RFUs analytical threshold analysis conditions to assess how the peaks may change from electrophoretic run to run and account for those variances when modeling stutter. At 50 RFUs, the variance factor was ~4.6728. A graph of the results is shown in Appendix A with the expected peak heights on the × axis plotted against the log of the heterozygous balance on the *y* axis. At 30 RFUs the variance factor was ~4.7881. A graph of the results, as described above, is shown in Appendix A. The relevant variance factor was linked to specific Protocol Sets, which were then selected as part of the MaSTR™ analysis procedure, ensuring that the correct values were used. The variance factors for each of the analytical thresholds used differed very little, indicating that the thresholds employed did not play a large role variance factor determination.

### 3.2. Number of MCMC Iterations and Precision

As described earlier, tests were run with different numbers of MCMC iterations to determine the most efficient parameters for analysis, as well as assess MaSTR™ software precision. Data were collected for LRs calculated using 40,000 and 80,000 total iterations for two-person mixtures, 80,000 and 160,000 total iterations for three-person mixtures, and 160,000 and 400,000 total iterations for the four- and five-person mixtures. Data from preliminary protocol development analyses with two- and three-person mixtures run with MCMC iteration totals up to 400,000 showed no improvement in results and longer run times; thus, only the lower totals for the two- and three-person mixtures stated above were pursued for this study.

Likelihood ratio data results from five replicate analyses were used to calculate mean values and standard deviations for the same contributors using the relevant different numbers of total iterations to determine if the iteration counts would yield practically significant differences. The results from a sub-set of mixture conditions are shown as bean plots in Figure 1a–f. These results are typical of those collected with MaSTR™ and are representative of the dataset used in this study. The LR proposition used for precision data collection was H_1_: one POI and N-1 unknown person(s) and H_2_: N unknown persons.

We performed Levene’s test to compare the variances of the results between different numbers of MCMC iterations for each sample. None of these tests were significant, with *p*-values > 0.05. We conclude that the length of the MCMC chain does not have a statistically significant impact on the variances of the LRs in this study. The two-person 1:1 undiluted (~500:500 pg) 1678/1653 mixture LRs (Figure 1a) showed that, while the number of MCMC iterations did not lead to different values using total values of 40,000 (5K/chain) or 80,000 (10K/chain) with eight chains, the standard deviation was not statistically smaller (Levene’s test *p*-value > 0.05) than with the analyses run with a total of 80,000 iterations. Similar results were shown with the three-person 1:1:1 undiluted (~500:500:500 pg) 1668/1658/1659 mixture (Figure 1b).

Figure 1c shows the LR results from the four-person 1:1:3:10 undiluted (~50:50:150:500 pg) 1682/1683/1690/1669 mixture using either 160K or 500K total iterations (20K or 50K per chain) and Figure 1d shows the LR data for the same mixture diluted 1:8 (~6:6:19:63 pg), again using 160K or 500K total iterations (20K or 50K per chain). Number of iteration results for the five-person 1:2:2:5:10 1665/1681/1686/1666/1670 undiluted (~50:100:100:250:500 pg) (Figure 1e) and diluted 1:8 (~6:13:13:31:63 pg) (Figure 1f) are also shown. Within both sets of contributor ratios, the 400,000 total iteration LRs have smaller standard deviations, though not statistically significant, (Levene’s test *p*-value > 0.05) than those analyses run at 160,000, except for contributor 1683 in the four-person 1:1:3:10 undiluted (~50:50:150:500 pg) mixture, which was an outlier.

The variance of LRs within each condition for major and minor contributors is within one log unit, except for the previously mentioned contributor 1683, which had one anomalous replicate test that yielded a value approximately three log units lower (9.65 × 10^3^) than the average of the other replicates (1.43 × 10^6^). It is of note that the same mixture sample analyzed using the lower MCMC iteration number showed less variance, but still had one outlier. Contributor 1683 was a minor contributor to this mixture, occupying the second position in the overall (sub-sub-source) LR calculations. Even in the undiluted sample, this contributor was present at less than 100 pg (Table 2) and displayed stochastic amplification characteristics as one allele dropped out in the D10S1248 locus (electropherogram not shown). Contributor 1683 had the genotype 13, 15 at this locus and the 15 allele was missing in the electropherogram used. Four of the five MaSTR™ analyses returned LRs of averaging 5.35 × 10^−2^ for this locus. The fifth analysis had an LR of 3.44 × 10^−5^, accounting for the outlier and demonstrating the effect that stochastic amplification can have on replicate LR values calculated for the same data. Despite these challenges, contributor 1683, as well as all the H_1_ true results tested for precision, still yielded LR values >1. Previous studies [44] using MCMC systems have also recorded LR values occasionally deviating by up to two log units and values within one log unit across different contributor ratios and peak heights is well within expectations for a precise system [45,46].

### 3.3. Accuracy Assessment

The accuracy of the MaSTR™ software was initially examined by comparing the results of manual calculations with those of the software. The single source known donor 1692 was chosen at random for this analysis. The random match probability (RMP) was determined for each locus using the Hardy–Weinberg equilibrium (HWE), and NRC 4.1 and NRC 4.2 recommended formulae using the Microsoft Excel spreadsheet and MaSTR™. The results demonstrated concordance between the values calculated in Excel and those from MaSTR™. These data are shown in Appendix A. The RMP of each of the true contributor genotypes used in the study was then calculated by hand and compared to the RMPs calculated by MaSTR™, each using the NRC 4.1 equation with a θ value of 0.01. The RMP values produced by hand and by MaSTR™ were concordant, with some minor differences due to rounding. The results of these calculations can be seen in Appendix A.

Manual calculations for individual loci in a relatively simple two-person mixture conditioned with a known minor contributor using the modeled weighted probabilities, including the possibility of drop-out (Q) alleles, from MaSTR™ were also performed and compared to the results for the same loci run with MaSTR™. The mixture chosen had a ratio of 1:10, making it relatively intuitive for an experienced analyst to identify the major contributor (1693) and deconvolute the mixture. The LRs calculated in each method were nearly identical. The details of the hand calculation results for the D2S441 locus are shown as an example in Appendix A. The LRs for all the loci calculated by hand using modeled weighted probabilities and by MaSTR™ were concordant, varying at only one locus (CSF1P0) with a value of 7.70 × 10^0^ by hand and 7.68 × 10^0^ with MaSTR™ (Appendix A). The complete profile LRs when calculated by hand or by MaSTR™ were also concordant at 2.43 × 10^28^ using modeled weighted probabilities (Appendix A). Additionally, each locus LR and the profile LR for only the unambiguous intuitive major contributor (no Q alleles) was calculated by hand using the unweighted probabilities (not modeled in MaSTR™) and a numerator of 1. The profile LR using weighted probabilities with Q alleles, whether by hand or MaSTR™, was 2.43 × 10^28^ while using the unweighted intuitive probabilities yielded a profile LR of 8.02 × 10^28^ (Appendix A).

The mixture modeling performance of MaSTR™ was examined using the same mixture combination described for the manual calculations (contributors 1660 and 1693) but using the full spread of ratios and DNA template amounts. The average LRs (N = 5) from the major contributor (1693) and minor contributor (1660) in mixtures with ratios of 1:1, 1:2, 1:3, 1:5, and 1:10 at each of the template quantities (Table 2) were plotted to see if the LR values followed the expected trends. If modeled correctly, the major contributor would be expected to show the lowest LR value at a mixture ratio of 1:1 (when there really is no major/minor contributor) and then its associated LRs should increase as the mixture ratio increases, delineating it more clearly from the minor as the difference in template DNA increases. The data for contributor 1693 conformed to the expected trend, with the lowest average LRs seen at the 1:1 ratio (1.57 × 10^14^ – 6.66 × 10^15^) and the highest in the 1:10 ratio (2.29 × 10^28^ – 1.08 × 10^29^) (Appendix A). The quantities of template for contributor 1693 were ~500 pg, ~125 pg, ~125 pg, and ~63 pg and did not vary by mixture ratio as described earlier. As expected, the average LR values increased with template quantity within a particular mixture ratio set. This was not always true, however, as the 1:10 ratio mixture had an average LR for 1693 at ~63 pg of 2.29 × 10^28^, while the average LR at ~125 pg was 9.56 × 10^26^ (Appendix A), indicating that stochastic amplification can still be a factor. The modeling performance of the minor contributor (1660) was also examined. The expected LR values for the major and minor contributors should be close to one another in the 1:1 ratio mixture (again, no real major/minor) and that was seen with average LRs for 1660 at 6.38 × 10^13^ – 4.67 × 10^16^ (Appendix A). The DNA template in the minor contributor decreases rapidly as the mixture ratios change, unlike those of the major contributor which stayed consistent throughout, due to the way the mixtures were built. Minor contributor template DNA ranged from ~500 pg down to ~6 pg and, if modeled correctly, LR values were expected to fall as well, especially in the 1:5 and 1:10 ratios. This was seen with average LRs in the 1:10 mixture ratio ranging from 4.14 × 10^2^ – 1.79 × 10^11^ (Appendix A)

Accuracy was further examined by performing a limited comparison of LRs generated by MaSTR™ to those of another validated probabilistic software package to examine concordance between the systems. The EuroForMix v3.3.1 (open access, http://www.euroformix.com/, accessed on 1 June 2022) software was selected, as it is open source, fully continuous, and validated for forensic probabilistic genotyping with mixtures of up to four contributors (20, 21). Mixture samples with two, three, and four contributors (including both two- and three-person high and low allele share constructs) were used for accuracy testing to determine if the two systems would return LRs that were similar. A range of undiluted mixture ratio profiles were examined. All tests used the same propositions: a POI genotype set as the reference contributor and N-1 unknown, unrelated genotypes set as alternates as the H_1_ proposition and N unknown genotypes as the H_2_ proposition. The “Quantitative LR (Bayesian based)” calculation in EuroForMix was selected for the analyses performed, as it corresponded most closely to the calculations used by MaSTR™. The resulting LRs from the two systems were generally concordant, and in some instances, the results were nearly the same (Appendix A). An exception was contributor 1683 in the three-person 1:1:1 (~500:500:500 pg) mixture. MaSTR™ yielded an LR of 9.55 × 10^3^, while EuroForMix gave 4.00 × 10^6^ (Appendix A). This was an undiluted mixture sample with equal amounts (~500 pg) of all contributors, so contributor quantity was high and no allele drop-out was observed. The ratio trace plot for this mixture met expectations for three equal contributors, indicating that the mixture was not skewed for any of the contributor genotypes. Contributor 1683 had 18 heterozygous loci out of a possible 22 and may possibly have come from a person who was highly genetically admixed. These data indicate that contributor 1683 may have been difficult to model in the mixture for both systems, considering the much higher LR values calculated for each of the other two mixture contributors by both programs. Two other contributors, 1669 and 1683, in the four-person 1:2:2:5 (~100:200:200:500 pg) ratio mixture had substantially higher LRs when calculated with MaSTR™ than with EuroForMix (Appendix A); however, each system still returned results that ranged from “Very Strong Support” to “Strong Support” on the verbal scale [47]. Neither software package demonstrated any false exclusions with the tested mixtures.

The MaSTR™ software produces a “Ratio Trace Plot” for each analysis run. This plot graphically represents the calculated amount of each contributor to the mixture based on peak heights. One example each of two-, three-, four-, and five person mixtures are shown in Figure 2 and are representative of the data generated in the larger set. All examples shown are from analyses that used no conditioning profiles. These plots indicated how closely the intended mixture ratios, as checked by hand during mixture creation using the GeneMarker^®^ HID v2.9.0 software, matched what MaSTR™ calculated from the relevant electropherogram data undergoing mixture deconvolution post burn-in. While minor deviations were expected due to amplification, electrophoretic, and detection variances, the desired ratios were closely reflected in the MaSTR™ plots, indicating that the dataset used was reliably providing the appropriate conditions. It should be noted that when MaSTR™ creates the “Ratio Trace Plots”, it defaults to setting the largest contributor as unknown 1 and then sequentially assigns numbers to contributors with decreasing amounts of detected signal, regardless of the actual order of reference or alternate contributors stipulated in the analysis set-up.

### 3.4. Testing Sensitivity and Specificity

Sensitivity and specificity, or the potential for the software to generate false exclusion and false exclusion errors, respectively [9], were examined, as well as the range of LR values that MaSTR™ generated with different contributor combinations. The software was challenged with a range of contributors, mixture ratios, peak heights, and stochastic amplification conditions representative of casework type samples. These included combinations where every true contributor allele was present at peak heights well above the Limit of Detection (LOD) set at either 30 or 50 RFUs, as previously described, to contributors with allele peak heights very near the LOD and extensive allele/locus drop-out.

Each time MaSTR™ performed an analysis run, it also generated 100 random contributor genotypes for each of the African American, Caucasian, Hispanic, and Asian allele frequency groups in the database chosen during the set-up process and calculated LRs for each genotype as a contributor to the mixture genotype being tested. It also generated 100 genotypes using allele frequencies taken at random from a combination of all the allele frequency groups and used those as contributors. This yielded 500 random in silico genotypes for use in each MaSTR™ run (regardless of the number of contributors in the mixture being tested). Every MaSTR™ analysis in this study was performed using five replicates, giving a total of 2500 results for in silico true non-contributors (expected LR < 1) data for every mixture sample run, in addition to the in vitro DNA true non-contributor test results. These provided a large number (~460,000 total for all MaSTR™ runs conducted in this study) of true non-contributor tests to augment those from the experimental in vitro DNA contributor data shown. The combination of in silico/in vitro true non-contributors provided a wide distribution to test against the mixture genotypes being analyzed and provided a range of LRs that could be expected in the MaSTR™ output. Figure 3 shows examples of in silico database LRs calculated for different mixture conditions including those with major and minor contributors and allele drop-out.

As the number of contributors and complexity of the undiluted mixtures increased, the LRs approached values near one, as expected. However, even the most complex undiluted mixture in our dataset, a five-person 1:2:2:5:10 (~50:100:100:250:500 pg) ratio sample that showed four total alleles dropped out, still demonstrated LRs < 1 ranging from 10^−21^ to 10^−2^ (Figure 3e) with randomly generated in silico non-contributor genotypes. Likelihood ratios ≥ 1 were most often observed for randomly created in silico non-contributor genotypes in the most highly diluted (1:8) complex mixtures that suffered from extensive allele drop-out, such as those shown in Figure 3b (35 alleles dropped out), Figure 3d (14 alleles dropped out), and Figure 3f (35 alleles dropped out). Across the five replicates run for each of these mixture types, there were 18 total LRs > 1 for random in silico non-contributor genotypes in the ~13:38:63 pg mixture, eight total for the ~6:6:19:63 pg mixture, and 21 total for the ~6:13:13:31:63 pg mixture. These data were typical of those seen when low template (diluted 1:4 or 1:8) mixtures containing minor contributors demonstrating stochastic amplification and allele drop-out were analyzed with the random in silico genotypes, yielding LRs ≥ 1 in 0.25–0.9% of the runs with those types of mixtures with MaSTR™.

The number of analyses for each mixture condition using in vitro DNA contributors are shown in Table 4. The number of tests listed equals the number of LRs calculated for that condition. Emphasis was placed on running H_1_ true tests with each condition, as this type of analysis would be expected to be most commonly encountered in casework.

All the data reported henceforth are from in vitro DNA contributors. The range of LRs corresponding to the verbal scale [47] for true contributors and non-contributors are shown in Figure 4. Each of the differing number of contributors’ data are shown in their own panel; however, the LR data from all the separate ratio, template quantity, and dilution conditions for each number of contributors have been combined to show unified ranges. As a general trend, the more complex the mixture sample—with more minor contributors and higher amounts of allele/locus drop-out—the more often LRs for true contributors were found to be close to, or below one (false exclusion). False inclusion errors were never observed under any in vitro DNA contributors for all numbers of contributors or DNA template conditions. False inclusion errors were seen in some of the random database results under extreme conditions of allele drop-out, as previously described (Figure 3).

The results for the two-person mixtures without any conditioning known contributors demonstrated LRs > 1 or LRs < 1 appropriate to the known contributor being tested (H_1_ or H_2_ true contributor) (Figure 4a). True contributors were nearly all (285/300) assigned LRs ≥ 10^6^, giving a result of “very strong support”, the highest on the verbal scale [47]. A small number (15/300) had LRs in the 10^2^ range and these values were from the replicate analyses of both the high and low contributor overlap combinations for the minor contributors in the 1:10 mixture ratios diluted at 1:4 (~13:125 pg) or 1:8 (~6:63 pg). These trace contributors (1660 and 1678, respectively) showed extensive allele drop-out, ranging from 12–21 missing alleles, depending on the tested condition. Lowered LR values for such low template contributors were expected and still represent “moderate support” on the verbal scale [47]. A limited sub-set of two-person H_1_ true mixtures were conditioned with a known contributor (Table 4). All LR values for two-person mixtures using a known contributor were ≥10^20^. No false exclusion errors were observed in any of the LRs calculated for in vitro DNA mixture analyses of two-person mixtures.

The results calculated for the three-person mixtures without any conditioning known contributors and including both true contributors and non-contributors demonstrated a broad range of LR values, including some observed false exclusion errors (Figure 4b). While more than 65% of the H_1_ true tests yielded LRs ≥ 10^6^, the remainder had a wide distribution of results of both LR > 1 and LR < 1 (Figure 4b). This could be explained by the increased complexity that this category entailed compared to the two-person mixtures. In addition to the innate complexity of adding contributors, the quantity of allelic information was shown as crucial. The lowest LR calculated for an H_1_ true contributor had an average value of 1.21 × 10^−4^, which corresponded to the minor contributor (1679) in a 1:3:5 mixture diluted 1:4 (~25:75:125 pg). This contributor had 17 alleles including four full loci drop-out. Within that same mixture, the middle contributor (1683) had two alleles drop-out and an average LR of 3.65 × 10^3^, demonstrating that low-level contributors could still yield LRs > 1, so long as the preponderance of alleles were detected. There were 38 H_1_ true tests that resulted in LRs < 1 (false exclusion error) out of a total of 538 three-person H_1_ true tests (Table 4) analyzed without a known contributor. All of these errors were assigned to minor or low-level contributors with missing alleles or loci such as described above. All tests with true non-contributor in vitro DNA genotypes on three-person mixtures resulted in LRs < 1. As with the two-person mixtures, a sub-set of the three-person mixtures were tested using a conditioning known contributor (Figure 4e). In all of these tests, the known contributor corresponded to the major or most abundant contributor, maintaining the most challenging parts of the mixtures for MaSTR™ to deal with. When conditioned, the percentage of H_1_ true LRs that were ≥10^6^ increased to 80% (Figure 4e). The remaining false exclusion errors (10/200 LR tests) were all from the two sets of replicate analyses of the 1:4 (~25:75:125 pg) and 1:8 (~13:38:63 pg) dilutions of contributor 1679 in the 1:3:5 mixture previously described. Additionally, contributor 1683 (middle position) in the same mixture sample (1:4 dilution) (~25:75:125 pg) had an average LR of 1.20 × 10^8^ when conditioned, an increase of nearly 5 log units from that described previously, indicating how substantially a conditioning known contributor can alter the test results.

The results calculated for the four-person mixtures without any known contributors and including both true contributors and non-contributors can be seen in Figure 4c. The range of LR values for the four-person mixtures was much narrower than that of the three- or five-person mixtures. Approximately 83% of the 420 unconditioned H_1_ true tests with four contributors resulted in LR values ≥10^6^ (Figure 4c). There were no false exclusion errors observed in any of the four-person H_1_ true tests, setting this category apart from the three- and five-person test results. This may have been due to none of the mixture samples displaying the extreme amounts of allele drop-out seen in the other mixture types, even though they were diluted to the same levels. Stochastic amplification was still evident in the four-person mixtures. The smallest LR calculated for a true contributor had an average value of 1.11 × 10^1^, which corresponded to one of the minor contributors (1683, second position) in a diluted 1:1:3:10 (~6:6:19:63 pg) mixture using ~6 pg of input DNA. This contributor had eight alleles, including one full locus, drop-out yet still yielded a moderately supportive LR on the verbal scale [47]. All true non-contributor genotypes in the four-person mixtures tested had LRs < 1 (Figure 4c). Due to the lack of false exclusion errors observed in the four-person results, only a small sub-set of the mixture samples was tested with a known contributor (Table 4). With this conditioning, the percentage of true contributors yielding LR values ≥10^6^ increased to 100%, while all true non-contributors had LRs ≤ 10^−6^. Additionally, as expected, the calculated LR values increased when the mixture was conditioned with a known contributor. An example is contributor 1682 (second position) in a 1:8 dilution of a 1:1:1:1 (~63:63:63:63 pg) mixture. This contributor had an average LR of 7.83 × 10^10^ with no conditioning known contributor and 1.41 × 10^12^ with one. This result is typical of the LR value increases seen in conditioned MaSTR™ calculations from mixture samples with little or no allele drop-out and, while not as large as that seen for contributor 1683 described above, still represents increases of ≥1 log unit.

Unconditioned LRs for both true contributors and non-contributors to the five-person mixtures can be seen in Figure 4d. Approximately 72% of the 730 unconditioned H_1_ true tests with five contributors resulted in LR values ≥10^6^ (Figure 4d). Similar to the three-person mixture results, there was a relatively wide distribution of LRs calculated for all types of contributors in this category. No false inclusion errors were seen with the in vitro DNA true non-contributors. There were 51 false exclusion errors observed within the five-person mixtures tested. These errors were all found in either minor contributors, diluted mixtures, or a combination of both and were not wholly unexpected as similar results have been seen in previous published work [21]. An example is contributor 1665 in a 1:1:5:5:10 mixture, occupying the first minor contributor position. This true contributor yielded LR < 1 values ranging from 1.03 × 10^−1^ (~6 pg, 15 alleles dropped out) to 5.72 × 10^−3^ (~13 pg, 10 alleles dropped out) depending on the specific dilution condition and level of stochastic amplification. These results were particularly interesting as they demonstrated how significantly amplification and electrophoretic separation and detection conditions can alter LR calculations. As described earlier, all mixtures were amplified in triplicate to assess peak height variance in the LR calculation process. In the 1:2 dilution of the 1:1:5:5:10 (~25:25:125:125:250 pg) mixture, contributor 1665 (~25 pg) had an average LR value of 2.32 × 10^3^ when the allele data from amplification one (three 1665 alleles dropped out) were used by MaSTR™. The average LR for 1665 changed to 8.99 × 10^−3^ when data from amplification two (seven 1665 alleles dropped out) were used. Data from amplification replicate number three yielded an average LR for 1665 of 7.39 × 10^6^ (four 1665 alleles dropped out). Of note is that the data for amplification replicate two had one full locus (D12S391, alleles 15 and 23) that dropped out, increasing the impact of the reduction in the LR over the other amplification replicates which only lost single alleles in heterozygous pairs. These data show a spread of LR values of approximately nine log units using the same contributor undergoing stochastic amplification. The LRs for the other contributors to this mixture, including the other minor (1666, ~25 pg), changed as well, showing LR value changes ranging from one to four log units, but they all remained greater than one. The electropherograms for the three replicates of this mixture are shown in Appendix A for comparison. Other true contributors in a variety of mixture conditions also demonstrated LR value shifts between replicate amplifications, some shifting from LRs < 1 to LRs > 1, but none to the extent that 1665 did in the mixture described. Representative examples of differences in LR values between amplification replicates of the same mixtures can be found in Appendix A. Some five-person mixture analyses were also conditioned with a known contributor. The results were very similar to those observed with the three-person mixtures. The largest contributor was again chosen as the known contributor and the percentage of conditioned sub-set tests yielding LRs ≥ 10^6^ was 50%; however, over 93% of the tests showed LRs > 1. The remaining false exclusion errors (5 total) were seen in the minor contributor (1665, first position in a 1:2:2:5:10 mixture, ~6:13:13:31:63 pg) in the most diluted mixtures with 13 alleles dropped out of the 1665 genotype.

### 3.5. Allele Peak Height, Allele Sharing, and Template Amount

The effects of the peak heights of only obligate (unshared) alleles in a mixture, overall contributor allele peak heights in a mixture, and the quantity of contributor DNA template for amplification on the average calculated LRs for low template (≤100 pg) true contributors was examined for the three-, four-, and five-person mixtures. It should be noted that the major contributors in some of these mixtures were amplified with DNA template quantities >100 pg, and thus, data for them were not included, while contributors that made up the “middle” positions in mixture combinations often were, as they fell below the 100 pg cutoff.

The average peak heights of only the obligate (unshared) alleles in each of the low template (≤100 pg) contributors were plotted versus the log10 of the average LRs of five replicate tests and the results are shown in Figure 5.

Most obligate alleles from all the mixture conditions clustered below 200 RFUs. The larger obligate allele peak heights came from those contributors that had DNA quantities of ≥60 pg and often occupied the middle positions of complex mixtures, such as the third position contributor (~75 pg) in a four-person 1:1:3:10 ratio that had been diluted 1:2 (~25:25:75:250 pg) (Table 2). These types of contributors were more prevalent in the four-person combinations than in the five-persons due to the quantities used to create the mixtures and pushed both the average peak heights and LRs of the four-person combinations relatively higher than the other mixture types.

The log10 of the average LRs of five replicate tests for individual contributors that were present in low template amounts (≤100 pg) in the mixtures described above were also examined (Appendix A). These data represent the range of LR values versus the average peak height in RFUs of all alleles, not just obligate alleles, detected for individual low-template contributors, regardless of allele sharing or specific mixture ratio, for three-, four-, and five-person mixtures. Data from the three-person mixtures includes examples of both high and low allele sharing mixture types. The LRs for the three-person high allele sharing mixtures tended to be greater than those of most of the three-person low share mixtures. As expected, the LRs for contributors present in smaller template amounts with correspondingly lower peak heights were less discriminatory, which agreed with the reported results from previous literature [48]. This generally held true for all the mixture types; however, there were some exceptions, as the data show that the relation between a low template contributor’s average peak height and its LR is not perfectly linear (Figure 5 and Appendix A). Stochastic amplification was observed in many of the genotypes of the contributors shown in the plot and allele drop-out was common in these types, as were large heterozygous peak imbalances. These variations were likely what caused some contributors that generally had higher average peak heights relative to some of the other contributors for the same type of mixture, to yield a calculated LR value that could be lower than expected, due to fewer alleles being available for modeling.

The quantities (pg) of individual low template contributors to the same mixtures shown in Figure 5 versus the log10 of the average LRs of five replicate tests were examined as well (Appendix A). The number of individual contributors with DNA quantities ≤40 pg was highest for the five-person mixtures with 18 shown. The prevalence of low template contributors in five-person combinations was again due to how the mixture quantities were put together and the number of minor or small contributors to the five-person ratios (Table 2). The individual contributors in the five-person mixtures also displayed more obligate alleles drop-out than those in the four-person mixtures with similar quantities. An example of greater drop-out impacting LR values can be seen in those contributors that were estimated to have had ~6 pg of DNA. Three of the minor contributors from the five-person mixtures with average LRs < 1 had very low average obligate peak heights and 7–12 obligate alleles dropped out (Appendix A). Very low template quantities did not always result in LRs < 1, however. The two contributors in the four-person mixtures that were estimated to have had ~6 pg of template DNA, obligate peak heights ranging from ~30–69 RFUs, and 5-6 obligate alleles dropped out still yielded LRs > 1 (Appendix A). Increasing the amount of template DNA for amplification in a mixture is not always a guarantee that calculated LRs will be >1 as long as stochastic effects continue to be applicable. Two individual contributors (1679 and 1670), one in a three-person and one in a five-person mixture, which had estimated quantities of ~60 pg, still resulted in average LRs that were < 1 (Appendix A). Contributor 1679 (first minor contributor) in a 1:1:2 ratio diluted 1:4 (~63:63:125 pg) (Table 2) had an average LR of 4.63 × 10^−2^ and an average obligate peak height of 87 RFUs. Contributor 1670 (third position contributor) in a 1:1:1:1:1 ratio diluted 1:8 (~63:63:63:63:63 pg) (Table 2) had an average LR of 5.98 × 10^−1^ and an average obligate peak height of 61 RFUs. Both contributors had some alleles drop out—four and two, respectively. However, neither was at the high end of drop out. These contributors, even though present in higher quantities than other low-level contributors, did not model into a specific positional order well, indicating that mixture samples affected by stochastic amplification can still present substantial challenges to probabilistic genotyping interpretation.

### 3.6. Number of Contributors

The number of contributors proposed to be in the tested mixture is one of the selectable parameters in the MaSTR™ set-up menu. To assess the impact of the number of contributors (NOC) on the calculated LR value, the three- and four-person mixtures were chosen to test one fewer (N-1) or one more (N + 1) contributor than the ground truth value. The three-person highly shared mixtures were selected for these tests to present MaSTR™ with the fewest alleles to work with and thus the most challenging situations. All mixture samples used for the NOC tests were undiluted. Tested conditions used a variety of propositions. One set of propositions included one genotype defined as the reference contributor, with the remaining contributors (−1) or (+1) defined as unknown, unrelated alternates in the numerator and either N − 1 or N + 1 unknown, unrelated profiles in the denominator with all genotypes used throughout being true contributors (Table 5 and Table 6). The N-1 contributor tests were also conditioned with one contributor genotype (minor, if present) identified as a known (Table 7) and the N + 1 contributor tests were conditioned with one genotype (major, if present) being a true non-contributor (Table 8). These conditions examined the effects on the LRs to yield false exclusions or inclusions when the wrong NOC was used. Average calculated LR results of the unconditioned N-1 contributor tests can be seen in Table 5.

Compared to the LR values for the true number of contributors, the N-1 LR values for either minor or equal contributor components of the mixture were almost always decreased by several log units, resulting in LRs < 1 (false exclusions) for true contributors. These types of reductions in the LRs have been documented previously [48,49] and were not unexpected. The reduction in LR values was not observed if the component contributor was a large or major contributor, such as 1659 in the three-person 1:2:10 (~50:100:500 pg) and 1:3:5 (~1000:300:500 pg) mixtures (Table 5). The average overall (sub-sub-source) LR values for the major contributors in both the unconditioned three- and four-person mixtures were very similar whether the N or N-1 NOC was selected. This was most likely due to the larger contributors’ increased peak heights and the ability to fully resolve their genotypes in the mixture, which is in agreement with other studies [48]. Considering that the overall (sub-sub-source) LR value calculation factors in the position of the putative contributor, as well as its presence, in the mixture, the average simple (sub-source) LR values were also examined to see how they would compare when contributor position was not included. The average simple (sub-source) LR values usually increased several log units for minor and/or equal contributors; however, they did not always rise high enough to prevent false exclusions. Major contributor average simple (sub-source) LRs showed increases as well but were less relevant as the values between different NOCs already demonstrated little difference (Table 5). The four-person 1:2:2:5 (~100:200:200:500 pg) mixture showed some atypical results, with contributor 1682 (second mixture position, ~200 pg) having a decreased, but still robust, LR value (1.10 × 10^13^) while its neighbors’ (1669 and 1683) LRs behaved as seen with other low level contributor values, dropping well into the <1 range (Table 5). None of the low-level contributors in this mixture demonstrated any allele drop-out, so the failure of contributor 1682 to yield a false inclusion when included in an analysis where the NOC was short by one indicated that MaSTR™ was able to successfully model this genotype into the mixture. Appendix A show the LRs for each contributor locus from one analysis replicate for the 1:2:2:5 (~100:200:200:500 pg) mixture described. 

Average calculated LR results of the conditioned N-1 contributor tests can be seen in Table 7. The minor contributor genotype (if present) was defined as a known, testing the ability of the MaSTR™ software to return an LR > 1 for true contributors in a mixture with too few NOCs identified. In mixtures with large major contributors (5X or 10X), the major genotype usually, but not always, yielded LRs > 1 (Table 7); however, the values were below those seen in tests using the ground truth NOCs. Low-level or additional minor contributors were usually forced into false exclusion LRs, although not always, as can be seen for contributor 1690 (~150 pg) in the four-person 1:1:3:10 mixture with an LR of 4.88 × 10^+1^. These results were, again, not unexpected, as there were simply too many contributor genotypes to model into the defined possible combinations. Averages of the simple (sub-source) LR values are also shown in Table 7 for comparison. As seen in the unconditioned N-1 results, the simple (sub-source) LRs for the four-person mixtures tended to be higher than the overall (sub-sub-source) LRs, as they do not factor in contributor order, just the presence of the genotype. Simple (sub-source) LR values for the large major contributors did not differ much from the overall (sub-sub-source) LR values, while some other minor/low contributors, such as 1690 (~150 pg, 1:1:3:10 mixture), did see increases. Of note is that the simple (sub-source) and overall (sub-sub-source) LR values for the three-person mixtures were the same. This was due to having a defined NOC of two, so the order of contributors is highly constrained as compared to mixtures with three or more contributors where more combinations were possible

Average calculated LR results of the unconditioned N + 1 contributor tests can be seen in Table 6. The LRs sometimes changed by several log units; however, these differences could be more or less than those seen for LRs calculated with the true NOCs (Table 6). There were no examples of false exclusions in the unconditioned tests using N + 1 contributors. As seen with the N-1 tests, genotypes that were present in large or major contributor positions demonstrated very little change in their reported LRs. Similar results have been described in other studies [48] and the data in the current work follow the same trends. Even minor contributors rarely showed changes of more than approximately two log units, although there were exceptions, such as contributor 1668 in the three-person 1:3:5 (~100:300:500 pg) mixture (Table 6). This minor contributor demonstrated stochastic amplification and allele drop-out and had an average LR six log units less when analyzed as a four-person mixture than as the ground truth three-person mixture.

Average calculated LR results of the conditioned N + 1 contributor tests can be seen in Table 8. The major contributor genotype (if present) was replaced with a random true non-contributor, testing the ability of the MaSTR™ software to return an LR < 1 for true exclusions in a mixture with too many NOCs identified, and therefore, providing additional potential allele combinations to fit the genotypes into the mixture. In the equal-ratio mixtures, one contributor was replaced at random with a non-contributor. MaSTR™ returned LRs < 1 for every combination tested, except one. The three-person 1:2:10 (~50:100:500 pg) mixture was run with the true non-contributor 1618 placed into the major (500 pg) position and resulted in an average LR = 1.28 × 10^0^ for 1618 (Table 8). While the remainder of the average LR values for the other true non-contributors were <1, they ranged for 1.88 × 10^−1^ to 1.89 × 10^−6^, values far higher than seen for the same genotypes in MaSTR™ runs using the correct NOC values (Table 8). The simple (sub-source) LR values for the conditioned N + 1 mixtures were also examined and followed the expected trend of yielding higher average LRs than those of the ordered overall (sub-sub-source) values (Table 8). The simple (sub-source) LRs for the true non-contributors were all >1, indicating that false inclusions were common when contributor ordering was not included in modeling the conditioned mixtures. When the correct NOC values were used, the true non-contributors’ simple (sub-source) LRs showed values far below one, similar to the overall (sub-sub-source) LR data seen in Table 8.

## 4. Discussion and Conclusions

This internal validation study tested the MaSTR™ software across a wide range of mixture conditions and propositions. Over 2600 tests were performed using two-, three-, four-, and five-person mixtures in a range of ratios that included examples of equal contributors to minor and major contributors. Tested conditions included examples of high and low levels of allele sharing, as well as allele and locus drop-out. MaSTR™ demonstrated that it could provide accurate and precise positive and negative match statistics for true contributors and non-contributors, respectively. The observed range of calculated LRs followed expected patterns observed for other probabilistic genotyping systems [21,44,50]. MaSTR™ yielded higher LR values for mixture contributors that had higher template and/or peak height quantities, as well as those contributors in the higher end of the contributor ratios.

The number of MCMC iterations needed to provide accurate results and explore precision in the MaSTR™ modeling was examined. Using a standard number of iterations for all conditions that will provide reliable results is certainly possible; however, the data gathered in this study indicated that for less complex mixtures, such as two- and three-persons, there are diminishing returns when using more than 200,000 iterations with MaSTR™. The primary benefit demonstrated was slightly less, though not significant, variation between replicate analyses, at the cost of more time used per analysis, as has been seen for other systems [23]. The actual run time will vary based on the specifications of the computer processing hardware and how many mixture profiles are queued up. During this work, analysis times using the highest iteration parameter value available ranged from approximately three minutes for a single two-person mixture analysis to approximately 30 min per analysis for five-person mixtures that had over twenty analyses queued to be processed. If time is not a factor, then using different numbers of iterations could be discarded in favor of a single, maximal value. Determining what works best is recommended as part of each laboratory’s internal validation experiments. There is no established way to ascertain a priori which number of iterations is ‘better’. Based on statistical theory, the longer chains should give more accurate LRs. However, we would recommend in practice that a post-hoc evaluation of convergence of the MCMC chain be performed [41]. The data presented in this study show that MaSTR™ can produce statistical data with precision that falls well within accepted values, even for minor contributors in complex mixtures.

The MaSTR™ software demonstrated sensitivity and specificity. The LRs were universally <1 for in vitro DNA true non-contributors using the true NOC value; however, some tests with a much larger number of in silico true non-contributor genotypes resulted in LRs > 1. These false inclusion errors occurred in mixtures of three or more persons with low DNA template contributors, which often showed allele/locus dropout. These results show that MaSTR™ performs similarly to other published studies regarding false inclusion errors [21]. A total of 104 false exclusion errors were observed with mixture samples out of 2663 tests performed. No false exclusion errors were observed in the two- and four-person mixture analyses. The majority (56) of these false exclusion errors were seen in minor contributors to the five-person mixtures with low DNA template values and where allele/locus drop-out were extensive. This was not unexpected, as mixtures with larger numbers of contributors that contain fewer alleles are less informative [21,48] and can be more prone to yielding LRs < 1 for true contributors under those conditions. It should be noted that replicate amplifications of these challenged samples resulted in differing genotypes for the contributors that fell in the stochastic range, and thus, differing LRs due to allele and locus drop-out. These LRs ranged from >1 to <1 (false exclusion error), and therefore, interpreting DNA profiles in the stochastic range, even with probabilistic software assistance, should be made with caution. Even with challenged samples, more than 65% of the total H_1_ true tests performed with MaSTR™ yielded LRs ≥ 10^6^ (very strong support) [47] and over 96% of the total H_1_ true tests resulted in LRs > 1. Allele peak heights, sharing, and minor contributor template quantities were also examined for those contributors whose total DNA in the mixture was ≤100 pg. The Promega PowerPlex^®^ Fusion 5C amplification kit has been shown to produce full or majority genotypes with low template and degraded DNA [51], so mixture contributors with template DNA inputs below 20 pg were still expected to yield some usable peak height information for MaSTR™. The data presented show that even at very low DNA template levels, in complex mixtures, H_1_ true contributors usually produced LRs > 1.

The number of contributors assigned to the mixture analysis is a critical parameter that can have profound influence on the calculated LRs [49,52,53,54,55,56]. There has been extensive work on how determining the NOC can best be accomplished [57,58,59,60,61,62,63,64]. While this study had the benefit of knowing the exact number of contributors in each mixture, that will rarely, if ever, be true in forensic casework. As such, three- and four-person mixtures were tested using one more or one fewer contributor than was ground truth. Data from the analyses with N-1 contributor usually resulted in overall (sub-sub-source) LRs < 1 for true contributors that were in equal ratio or small/minor components of the mixture. Large or major contributor LRs usually resulted in little practical change with N-1 contributors identified. In some casework scenarios, there may be limited a priori knowledge of a person being in a mixture, such as a minor female victim profile in the sperm-rich fraction of a vaginal swab. In such a situation, the LR for the minor contributor may be <1 and could indicate that the wrong number of contributors has been chosen; however, most of the time, this will not be the case. When conditioned with a known contributor to the mixture, the N-1 LRs almost all were <1 except for those for major contributors, demonstrating that false exclusions could be forced if the NOC value was short by one.

When the same analyses were conducted using one more contributor than the ground truth and unconditioned, the results were less dramatic, especially for the large/major components, and consistent with other studies [48]. Equal and small/minor contributors showed both increases and decreases in their overall (sub-sub-source) LRs when N + 1 contributors were used, but true contributor LRs never fell into the <1 range and no false exclusions were observed. When the mixtures were run with N + 1 contributors and conditioned, one true non-contributor genotype did yield LRs > 1. This result demonstrated that false inclusions could also be forced if the NOC value was one too many. Together, these results demonstrate that careful attention should be given to determining the appropriate number of contributors to select when setting up a MaSTR™ analysis. Considering that multiple propositions are becoming a more common feature of forensic statistical analysis [65], perhaps performing multiple runs using differing numbers of contributors could be useful for mixtures with low/minor components, especially in the stochastic amplification range.

MaSTR™ was also demonstrated to provide accurate statistical information. Hand calculations for individual genotype RMPs, as well as locus and profile LRs for a relatively simple two-person mixture, were performed and compared to the same calculations done by MaSTR™. Concordant results were seen in all of those tests. In direct comparison with EuroForMix, another fully continuous probabilistic system, the LRs generated by the two software packages using the same propositions were generally concordant. Some variation in the LR results between the two was expected, as different software systems will yield somewhat different values for the same contributor [66]; however, the variations tended to be relatively small and rarely changed the outcome on the verbal scale. While this study included diluted mixture samples to examine the effects of stochastic amplification on the calculated LR values, it did not address degraded DNA in mixtures. That work has been examined in detail by another research group [67] to further demonstrate that MaSTR™ is an appropriate tool for forensic DNA analysis. MaSTR™ is also capable of performing analyses including kinship; however, that subject was not considered in this work, as it needs an in-depth analysis suited to a separate study and is the topic for future work. This study exclusively reported the LR results using only the All Dataset allele frequencies from the Hill et al. [35] database, and thus, was a limitation of the study. MaSTR™ also calculated all LRs using the additional population allele frequencies, which were different depending on which population was used; however, including this information was not part of the aims of the study. Casework laboratories performing their own validation will need to address which population allele frequencies are used and how they are reported out.

The data presented from the tested conditions indicate that the MaSTR™ software is accurate, precise, sensitive, and reliable for mixed profile STR statistical analysis. In experiments covering the requirements for validation of a probabilistic genotyping system, it has performed in accordance with expectations and in similar ways to other validated systems.

## Figures and Tables

**Figure 1 genes-13-01429-f001:**
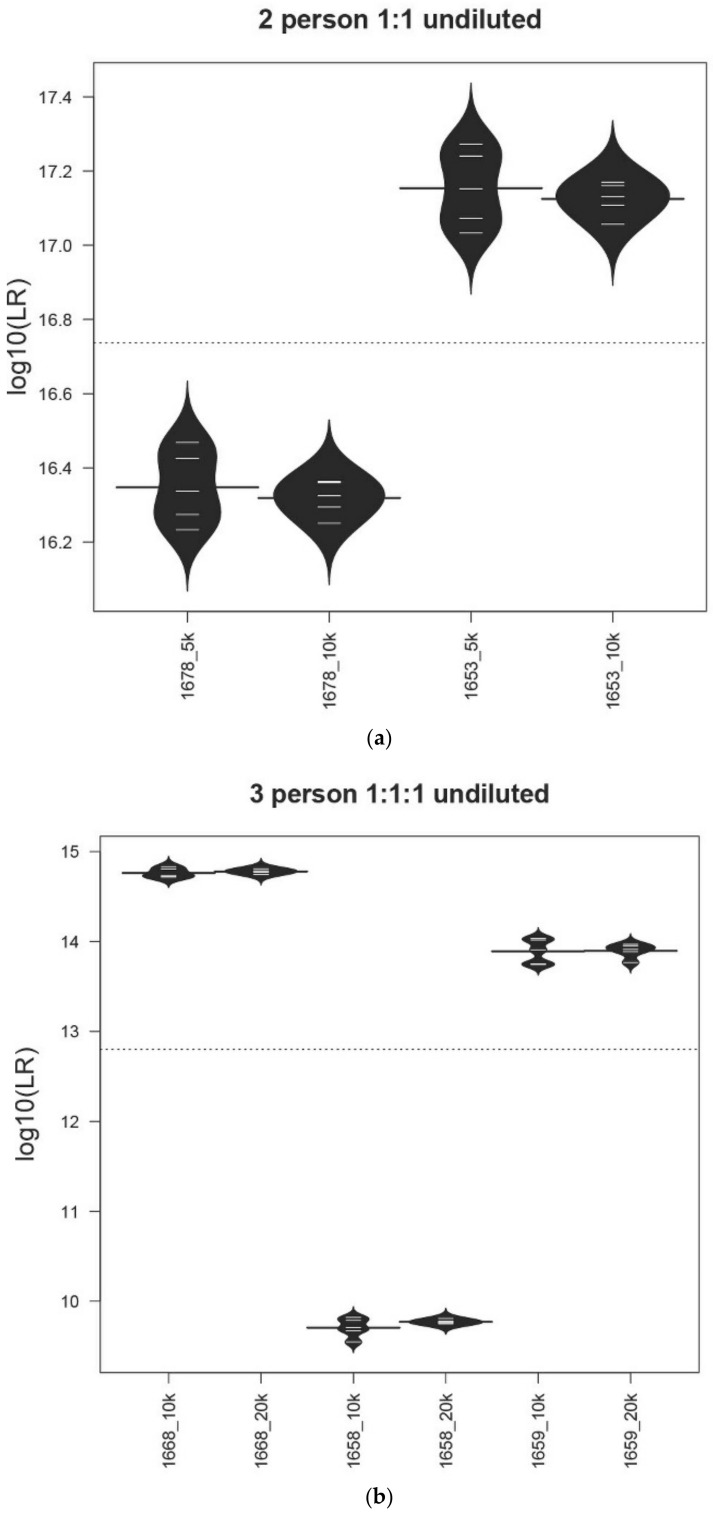
Precision in the MaSTR™ system. Bean plot representation of the average LRs and associated standard deviations for each contributor in a range of mixture ratio and DNA template quantity conditions. The average (N = 5) LR values for each contributor in the analyses using the different number of MCMC iterations per chain are shown for comparison (log10of the LRs on *y*-axis). (**a**) 5K/chain or 10K/chain for two-person (~500:500 pg), (**b**) 10K/chain or 20K/chain for three-person (~500:500:500 pg), (**c**) 20K/chain or 50K/chain for four-person (~50:50:150:500 pg), (**d**) 20K/chain or 50k/chain for four-person (~6:6:19:63 pg), (**e**) 20K/chain or 50K/chain for five-person (~50:100:100:250:500 pg), and (**f**) 20K/chain or 50K/chain for five-person mixtures (~6:13:13:31:63 pg). The dotted line indicates the overall mean of all data in the plot.

**Figure 2 genes-13-01429-f002:**
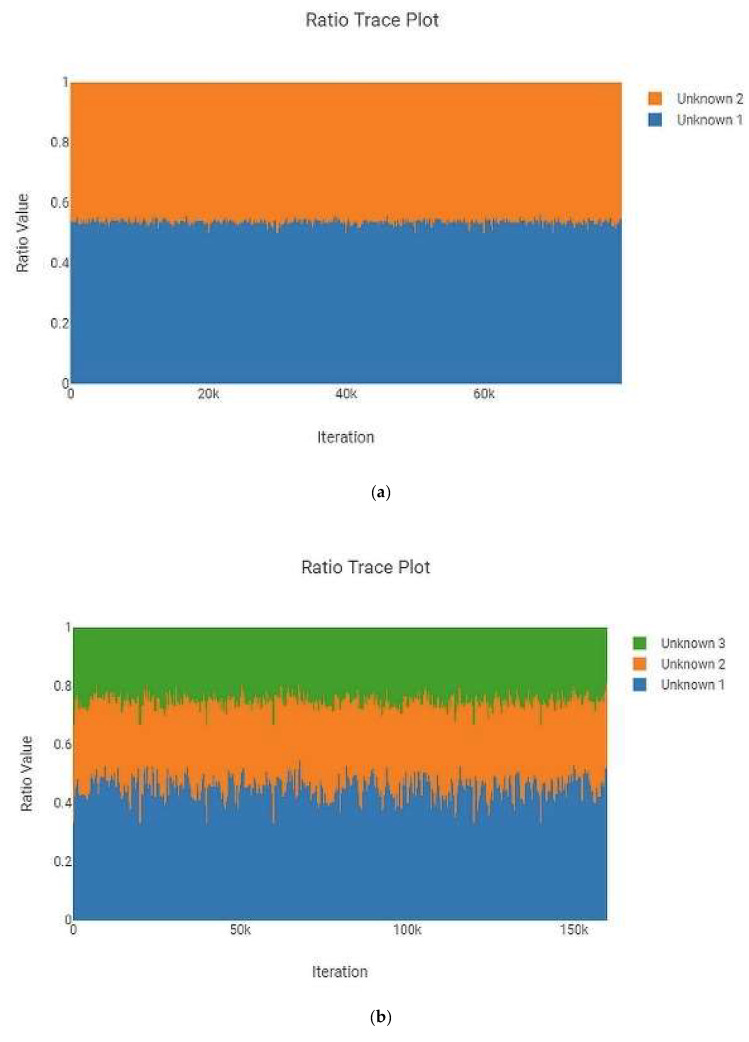
MaSTR™-generated ratio trace plots show a graphical representation of the calculated amount of each contributor to the mixture using the peak heights as determined from the modeled combinations of alleles in the most likely genotypes. The software defaults to numbering the unknown contributors (reference and alternates) from most abundant to least. The mixture ratios shown are (**a**) two-person 1:1, (**b**) three-person 1:1:2, (**c**) four-person 1:1:3:10, and (**d**) five-person 1:1:5:5:10. All data shown are from analyses using the highest number of MCMC iterations for the number of contributors.

**Figure 3 genes-13-01429-f003:**
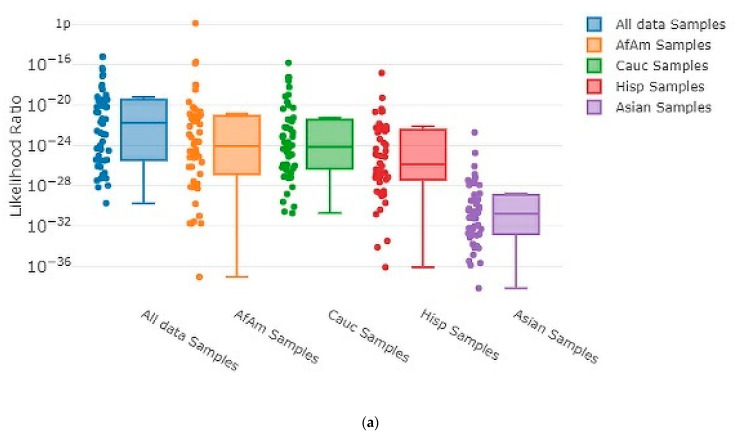
MaSTR™-generated random database in silico contributor genotype LR plots. MaSTR™ synthesized 100 random genotypes using allele frequency data from the four groups shown, as well as 100 additional genotypes synthesized from a combination of the four to yield 500 random genotypes for testing with each analysis run performed. The mixture ratios and template quantities shown are (**a**) three-person 1:3:5 (~100:300:500 pg), (**b**) three-person 1:3:5 (~13:38:63 pg), (**c**) four-person 1:1:3:10 (~50:50:150:500 pg), (**d**) four-person 1:1:3:10 (~6:6:19:63 pg), (**e**) five-person 1:2:2:5:10 (~50:100:100:250:500 pg), and (**f**) five-person 1:2:2:5:10 (~6:13:13:31:63 pg). All data shown are from analyses using the highest number of MCMC iterations for the number of contributors. The chart value “f” indicates 10^−15^, “p” indicates 10^−12^, and “µ” indicates 10^−6^.

**Figure 4 genes-13-01429-f004:**
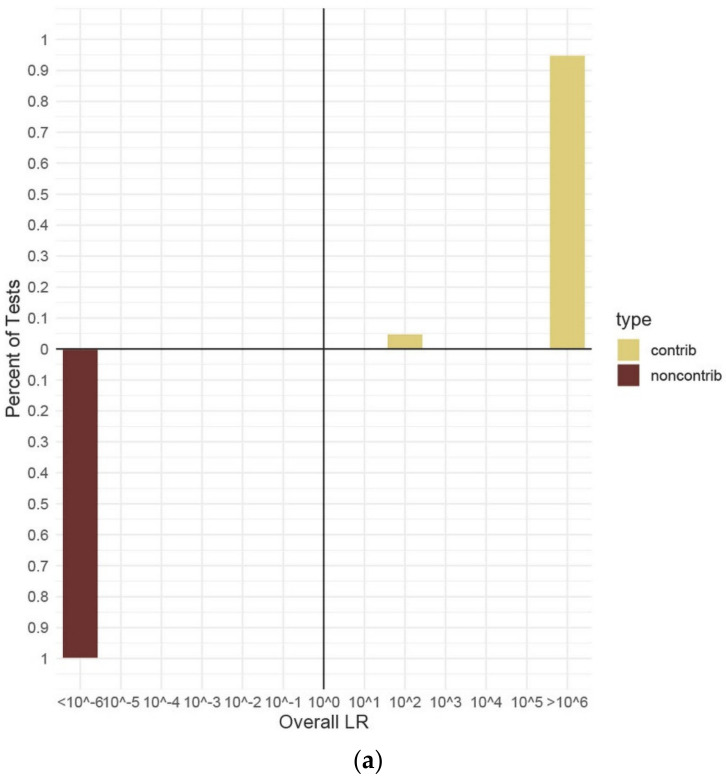
Percentage of overall (sub-sub-source) LRs for true contributors and non-contributors. (**a**). Two-person mixtures. (**b**). Three-person mixtures. (**c**). Four-person mixtures. (**d**). Five-person mixture€(**e**). Three-person mixtures conditioned with defined known contributor. Panel (**a**) shows two-person data from 300 H_1_ true and 40 H_2_ true tests with no conditioning knowns. Panel (**b**) shows three-person data from 538 H_1_ true and 90 H_2_ true tests with no conditioning knowns. Panel (**c**) shows four-person data from 420 H_1_ true and 60 H_2_ true tests with no conditioning knowns. Panel (**d**) shows five-person data from 730 H_1_ true and 95 H_2_ true tests with no conditioning knowns. €el (**e**) shows three-person data from 200 H_1_ true tests with a conditioning known identified. All data are from in vitro DNA samples analyzed using the range of hypotheses listed in Table 3 and the maximum number of MCMC iterations for the number of contributors. An LR value of ≥10^6^ represents “Very Strong Support” on the verbal scale [47], and so, all results meeting this threshold were combined on the charts.

**Figure 5 genes-13-01429-f005:**
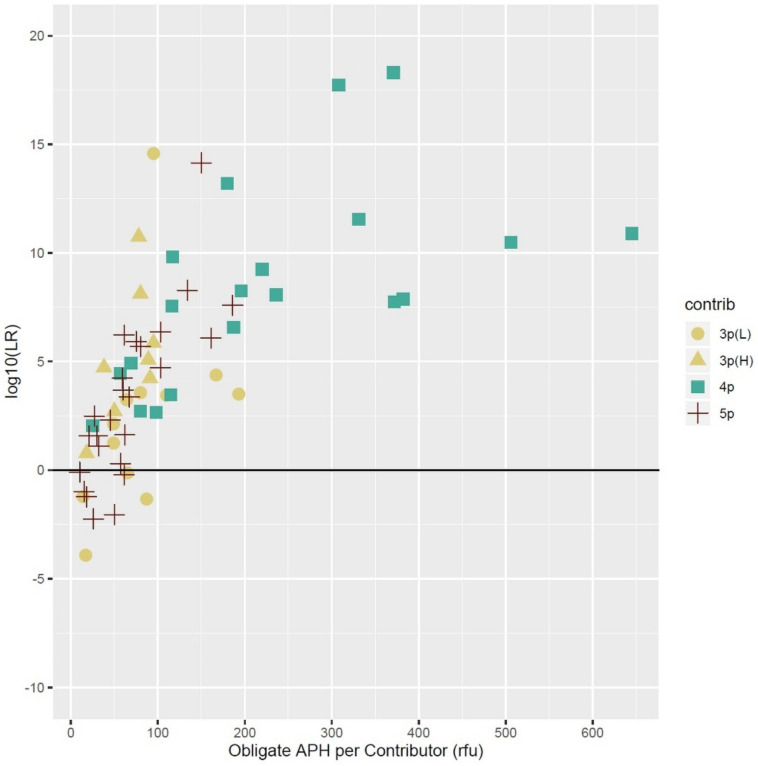
The log10 of the average LRs (N = 5) versus obligate average peak height (APH) for individual contributors that were present in low template amounts (≤100 pg). Each data point represents the obligate average peak height in RFUs representing all unshared alleles detected in a mixture for selected individual low-template contributors, regardless of mixture ratio. In total, 20–25 values each were selected for the three-, four-, and five-contributor mixtures that demonstrated the range of LRs observed in the study.

**Table 1 genes-13-01429-t001:** Overview of the combinations of single source contributors in each mixture type.

Type of Mixture	Contributor Code				
	Contributor 1	Contributor 2	Contributor 3	Contributor 4	Contributor 5
Low share, 2 person	1678	1653	N/A	N/A	N/A
High share, 2 person	1660	1693	N/A	N/A	N/A
Low share, 3 person	1679	1683	1657	N/A	N/A
High share, 3 person	1668	1658	1659	N/A	N/A
Random, 4 person	1669	1682	1683	1690	N/A
Random, 5 person	1665	1666	1670	1681	1686

**Table 2 genes-13-01429-t002:** Summary of mixture ratios, quantities, proportions, and dilutions analyzed with the MaSTR™ software. The approximate undiluted contributor DNA template in picograms indicates the quantity of DNA used to create the base (undiluted) mixture for each ratio. These mixtures were then serially diluted as indicated. The approximate quantity in picograms of specific contributors described in the paper is listed in the text where relevant. The approximate undiluted total picograms DNA template indicates all contributor template quantities added together for each of the ratios used. The total number of mixture conditions tested is the product of the number of ratios multiplied by the number of dilutions used (including undiluted). The number of mixtures for interpretation values reflect that each mixture condition was amplified in triplicate, yielding different peak heights for analysis.

Number of Contributors	Contributor Ratios	Approximate Undiluted Contributor DNA Template in Picograms	Approximate undiluted Total Picograms DNA Template	Sample Serial Dilutions	Total Number of Mixture Conditions Tested	Number of Mixtures for Interpretation
2 allele low share	1:1, 1:2, 1:3, 1:5, 1:10	500:500, 250:500, 160:500, 100:500, 50:500	1000, 750, 660, 600, 550	1:2, 1:4, 1:8	20	60
2 allele high share	1:1, 1:2, 1:3, 1:5, 1:10	500:500, 250:500, 160:500, 100:500, 50:500	1000, 750, 660, 600, 550	1:2, 1:4, 1:8	20	60
3 allele low share	1:1:1, 1:1:2, 1:2:10, 1:3:5	500:500:500, 250:250:500, 50:100:500, 100:300:500	1500, 1000, 650, 900	1:2, 1:4, 1:8	16	48
3 allele high share	1:1:1, 1:1:2, 1:2:10, 1:3:5	500:500:500, 250:250:500, 50:100:500, 100:300:500	1500, 1000, 650, 900	1:2, 1:4, 1:8	16	48
4	1:1:1:1:1, 1:1:3:10, 1:2:2:5	500:500:500:500, 50:50:150:500, 100:200:200:500	2000, 750, 1000	1:2, 1:4, 1:8	12	36
5	1:1:1:1:1, 1:1:5:5:10, 1:2:2:5:10	500:500:500:500:500, 50:50:250:250:500, 50:100:100:250:500	2500, 1100, 1000	1:2, 1:4, 1:8	12	36

**Table 3 genes-13-01429-t003:** Summary of the propositions tested for each mixture type with the MaSTR™ software. In MaSTR™, the person of interest (POI) genotype is defined as a “reference” and additional unknown genotypes are defined as “alternates”. MaSTR™ would not calculate an absolute exclusion (LR = 0) for a defined contributor; it assigned LR values < 1 with no lower limit for true non-contributors.

Number of Contributors	H_1_/H_2_ Propositions	H_1_/H_2_ Propositions
2 Person	1 POI and 1 Unknown *	1 Known and 1 POI
	2 Unknowns	1 Known and 1 POI
3 Person	1 POI and 2 Unknowns *	1 Known and 1 POI and 1 Unknown
	3 Unknowns	1 Known and 2 Unknowns
4 Person	1 POI and 3 Unknowns *	1 Known and 1 POI and 2 Unknowns *
	4 Unknowns	1 Known and 3 Unknowns
5 Person	1 POI and 4 Unknowns *	1 Known and 1 POI and 3 Unknowns
	5 unknowns	1 Known and 4 Unknowns

* Unknown genotypes were composed of both true contributors and true non-contributors, depending on the condition and analysis being conducted.

**Table 4 genes-13-01429-t004:** Summary of the number of analyses and specific condition tests for each mixture type performed with the MaSTR™ software. The total number of tests was determined by multiplying the number of analyses for a given mixture type by the number of contributors, i.e., each two-person mixture analysis will yield two LRs, each three-person mixture will yield three LRs, etc. Conditioned tests had at least one true contributor defined as a known. The totals shown here are only for tests performed using the correct number of contributors (NOC).

Number of Contributors	Total Number of MaSTR™ Analyses	Number of H_1_ True Tests with No Conditioning	Number of H_1_ True Tests Conditioned with a Known Contributor	Number of H_2_ True Tests with No Conditioning	Number of H_2_ True Tests Conditioned with a Known Contributor
2	195	300	50	40	0
3	276	538	200	90	0
4	135	420	40	60	20
5	181	730	80	95	0

**Table 5 genes-13-01429-t005:** Summary of the average (N = 5) calculated LR results of the N-1 NOC tests for each of the component contributors in its respective mixture. The contributors in each mixture are listed in the order of the position and ratio that they occupied in the mixture. The four-person mixtures altered the sample order, depending on the ratio being tested, for additional variability. All data shown are from undiluted samples, with the approximate quantities of DNA template in each mixture indicated. Overall (sub-sub-source) LRs consider the position of the individual contributor in the mixture, while simple (sub-source) LRs do not. The average overall (sub-sub-source) LR values for each mixture contributor calculated with the true NOC is included for comparison.

Number of True Contributors	Mixture Ratio (Mixture Quantities)	Contributor	N-1 True Contributor Overall Ave. LR	N True Contributor Overall Ave. LR	N-1 True Contributor Simple Ave. LR
3	1:1:1 (~500:500:500 pg)	1668	4.94 × 10^−1^	1.76 × 10^12^	4.97 × 10^5^
		1658	1.18 × 10^−23^	5.09 × 10^15^	1.64 × 10^−14^
		1659	1.67 × 10^−19^	2.62 × 10^10^	1.58 × 10^−3^
3	1:1:2 (~250:250:500 pg)	1668	8.87 × 10^−13^	3.18 × 10^13^	1.11 × 10^−3^
		1658	3.93 × 10^−20^	4.14 × 10^11^	1.50 × 10^−16^
		1659	3.85 × 10^6^	1.67 × 10^21^	1.76 × 10^12^
3	1:2:10 (~50:100:500 pg)	1668	2.46 × 10^−16^	9.7 × 10^4^	8.87 × 10^−14^
		1658	1.02 × 10^3^	1.67 × 10^12^	3.94 × 10^5^
		1659	3.55 × 10^30^	2.27 × 10^30^	4.18 × 10^31^
3	1:3:5 (~100:300:500 pg)	1668	3.59 × 10^−23^	1.56 × 10^4^	3.67 × 10^−13^
		1658	4.21 × 10^11^	7.85 × 10^10^	3.94 × 10^18^
		1659	2.65 × 10^26^	2.89 × 10^21^	4.18 × 10^27^
4	1:1:1:1 (~500:500:500:500 pg)	1669	2.08 × 10^−9^	8.00 × 10^9^	8.89 × 10^15^
		1682	9.77 × 10^−9^	1.32 × 10^12^	3.16 × 10^17^
		1683	4.38 × 10^−2^	5.16 × 10^10^	1.27 × 10^15^
		1690	1.76 × 10^−12^	1.19 × 10^7^	1.56 × 10^3^
4	1:1:3:10 (~50:50:150:500 pg)	1682	1.70 × 10^−5^	4.4 × 10^10^	2.03 × 10^6^
		1683	8.64 × 10^−12^	1.15 × 10^6^	1.09 × 10^2^
		1690	3.45 × 10^18^	7.61 × 10^21^	1.85 × 10^22^
		1669	1.26 × 01^28^	7.77 × 10^27^	1.44 × 10^30^
4	1:2:2:5 (~100:200:200:500 pg)	1669	8.15 × 10^−18^	8.98 × 10^11^	1.23 × 10^5^
		1682	1.10 × 10^13^	7.27 × 10^17^	3.29 × 10^19^
		1683	9.05 × 10^−6^	5.32 × 10^10^	2.36 × 10^6^
		1690	1.63 × 10^25^	4.89 × 10^25^	1.55 × 10^28^

**Table 6 genes-13-01429-t006:** Summary of the average (N = 5) calculated LR results of the N + 1 NOC tests for each of the component samples in its respective mixture. The contributors are listed in the order of the position and ratio that they occupied in the mixture. The four-person mixtures altered the sample order, depending on the ratio being tested, for additional variability. All data shown are from undiluted samples, with the approximate quantities of DNA template in each mixture indicated. Overall (sub-sub-source) LRs consider the position of the individual contributor in the mixture, while simple (sub-source) LRs do not. The average overall (sub-sub-source) LR values for each mixture contributor calculated with the true NOC is included for comparison.

Number of True Contributors	Mixture Ratio	Sample	N + 1 True Contributor Overall Ave. LR	N True Contributor Overall Ave. LR
3	1:1:1 (~500:500:500 pg)	1668	1.78 × 10^13^	1.76 × 10^12^
		1658	1.89 × 10^10^	5.09 × 10^15^
		1659	4.81 × 10^14^	2.62 × 10^10^
3	1:1:2 (~250:250:500 pg)	1668	7.13 × 10^13^	3.18 × 10^13^
		1658	7.10 × 10^10^	4.14 × 10^11^
		1659	2.31 × 10^21^	1.67 × 10^21^
3	1:2:10 (~50:100:500 pg)	1668	1.01 × 10^5^	9.76 × 10^4^
		1658	8.28 × 10^11^	1.67 × 10^12^
		1659	1.67 × 10^30^	2.27 × 10^30^
3	1:3:5 (~100:300:500 pg)	1668	3.35 × 10^4^	7.85 × 10^10^
		1658	9.55 × 10^11^	1.56 × 10^4^
		1659	6.11 × 10^23^	2.89 × 10^21^
4	1:1:1:1 (~500:500:500:500pg)	1669	4.51 × 10^11^	8.00 × 10^9^
		1682	3.70 × 10^12^	1.32 × 10^12^
		1683	3.95 × 10^10^	5.16 × 10^10^
		1690	5.60 × 10^8^	1.19 × 10^7^
4	1:1:3:10 (~50:50:150:500 pg)	1682	1.66 × 10^10^	4.40 × 10^10^
		1683	3.73 × 10^5^	1.15 × 10^6^
		1690	3.63 × 10^21^	7.61 × 10^21^
		1669	5.54 × 10^27^	7.77 × 10^27^
4	1:2:2:5 (~100:200:200:500 pg)	1669	5.15 × 10^9^	8.98 × 10^11^
		1682	5.98 × 10^14^	7.27 × 10^17^
		1683	9.47 × 10^8^	5.32 × 10^10^
		1690	4.93 x10^24^	4.89 × 10^25^

**Table 7 genes-13-01429-t007:** Summary of the average (N = 5) calculated LR results conditioned with a known of the N-1 NOC tests for each of the component contributors in its respective mixture. The contributors in each mixture are listed in the order of the position and ratio that they occupied in the mixture. The four-person mixtures altered the sample order, depending on the ratio being tested, for additional variability. All data shown are from undiluted samples, with the approximate quantities of DNA template in each mixture indicated. Overall (sub-sub-source) LRs consider the position of the individual contributor in the mixture, while simple (sub-source) LRs do not.

Number of True Contributors	Mixture Ratio	Sample	N-1 True Contributor Overall Ave. LR Conditioned w/Known	N-1 True Contributor Simple Ave. LR Conditioned w/Known
3	1:1:1 (~500:500:500 pg)	1668	Known	Known
		1658	4.01 × 10^−12^	4.01 × 10^−12^
		1659	2.45 × 10^−26^	2.45 × 10^−26^
3	1:1:2 (~250:250:500 pg)	1668	Known	Known
		1658	5.36 × 10^−24^	5.36 × 10^−24^
		1659	4.18 × 10^−4^	4.18 × 10^−4^
3	1:2:10 (~50:100:500 pg)	1668	Known	Known
		1658	4.87 × 10^−27^	4.87 × 10^−27^
		1659	1.57 × 10^1^	1.57 × 10^1^
3	1:3:5 (~100:300:500 pg)	1668	Known	Known
		1658	8.67 × 10^−18^	8.67 × 10^−18^
		1659	1.95 × 10^−4^	1.95 × 10^−4^
4	1:1:1:1 (~500:500:500:500pg)	1669	Known	Known
		1682	6.95 × 10^−18^	1.86 × 10^3^
		1683	4.26 × 10^−25^	4.05 × 10^−7^
		1690	6.44 × 10^−24^	2.49 × 10^−7^
4	1:1:3:10 (~50:50:150:500 pg)	1682	Known	Known
		1683	3.33 × 10^−25^	1.17 × 10^−17^
		1690	4.88 × 10^1^	5.61 × 10^5^
		1669	8.52 × 10^26^	2.01 × 10^28^
4	1:2:2:5 (~100:200:200:500 pg)	1669	Known	Known
		1682	6.78 × 10^−10^	1.80 × 10^−2^
		1683	1.45 × 10^−29^	1.13 × 10^−21^
		1690	2.51 × 10^23^	5.20 × 10^+23^

**Table 8 genes-13-01429-t008:** Summary of the average (N = 5) calculated LR results of the N + 1 NOC tests, including a true non-contributor (*) for each of the component samples in its respective mixture. The contributors in each mixture are listed in the order of the position and ratio that they occupied in the mixture. The four-person mixtures altered the sample order, depending on the ratio being tested, for additional variability. All data shown are from undiluted samples, with the approximate quantities of DNA template in each mixture indicated. Overall (sub-sub-source) LRs consider the position of the individual contributor in the mixture, while simple (sub-source) LRs do not. The average overall (sub-sub-source) LR values for each mixture contributor calculated with the true NOC is included for comparison.

Number of True Contributors	Mixture Ratio	Sample	N + 1 Contributor Overall Ave. LR	N + 1 Contributor Simple Ave. LR	True NOC Overall Ave. LR
3	1:1:1 (~500:500:500 pg)	1668	1.88 × 10^13^	2.00 × 10^27^	9.49 × 10^11^
		1658	1.95 × 10^10^	6.30 × 10^22^	1.53 × 10^10^
		1637 *	3.50 × 10^−3^	1.19 × 10^4^	1.79 × 10^−26^
3	1:1:2 (~250:250:500 pg)	1668	7.30 × 10^13^	3.88 × 10^25^	3.49 × 10^13^
		1658	5.19 × 10^10^	1.03 × 10^21^	5.80 × 10^11^
		1651 *	3.11 × 10^−1^	6.23 × 10^7^	5.49 × 10^−19^
3	1:2:10 (~50:100:500 pg)	1668	1.26 × 10^5^	9.26 × 10^15^	1.11 × 10^5^
		1658	8.35 × 10^11^	1.24 × 10^21^	1.79 × 10^12^
		1618 *	1.28 × 10^00^	3.46 × 10^10^	1.13 × 10^−4^
3	1:3:5 (~100:300:500 pg)	1668	4.55 × 10^4^	2.05 × 10^16^	5.05 × 10^5^
		1658	7.01 × 10^11^	8.45 × 10^21^	6.14 × 10^12^
		1687 *	8.43 × 10^−2^	2.74 × 10^5^	2.29 × 10^−16^
4	1:1:1:1 (~500:500:500:500pg)	1669	3.97 × 10^11^	7.47 × 10^25^	3.09 × 10^10^
		1682	3.74 × 10^12^	1.60 × 10^26^	1.50 × 10^12^
		1683	3.77 × 10^10^	2.98 × 10^24^	4.34 × 10^10^
		1689 *	1.88 × 10^−1^	2.87 × 10^8^	3.43 × 10^−22^
4	1:1:3:10 (~50:50:150:500 pg)	1682	1.67 × 10^10^	9.06 × 10^22^	4.75 × 10^10^
		1683	1.03 × 10^6^	1.67 × 10^20^	1.16 × 10^5^
		1690	2.95 × 10^21^	1.56 × 10^29^	6.74 × 10^21^
		1625 *	2.89 × 10^−2^	9.82 × 10^8^	1.22 × 10^−17^
4	1:2:2:5 (~100:200:200:500 pg)	1669	6.33 × 10^9^	1.27 × 10^23^	1.08 × 10^12^
		1682	1.25 × 10^15^	3.11 × 10^26^	7.16 × 10^17^
		1683	1.87 × 10^9^	3.31 × 10^22^	4.58 × 10^10^
		1677 *	1.89 × 10^−6^	1.96 × 10^4^	5.49 × 10^−17^

## Data Availability

Not applicable.

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
