# Peer review of "Internal Validation of MaSTR™ Probabilistic Genotyping Software for the Interpretation of 2–5 Person Mixed DNA Profiles"

_genes, 2022, doi:10.3390/genes13081429_

Round 1

Reviewer 1 Report

This article describes and internal validation of the MaSTR probabilistic genotyping platform using the PowerPlex Fusion STR multiplex. The authors have done a very thorough job in analyzing the data generated; however, I have several major and minor comments that I think will help improve the work itself and their presentation of the findings. My two major comments relate to (i) manuscript length and (ii) the selection of subsets of data to perform certain analyses that prohibit statistical comparisons. I appreciate the thorough description of all of the data but it might be easier for future readers of the work to have a select few highlights in each section. I recommend moving some of the material to a supplementary results section, if possible. I elaborate below on areas where I think statistical comparison of the data could be beneficial to support the authors message. Additional comments and critiques are provided below.

Introduction: please remove the indication that semi- and fully-continuous software needs to validated for “factually accurate” results. The goal of probabilistic genotyping is not to determine factual versus false results.

Section 2.1 – is there a reference for the Nebraska Biobank that could be cited here?

Section 2.3 – is there a reference for the 30rfu internally validated analytic threshold?

Section 2.3 – the rationale for using a higher AT for 2 and 3 person mixtures is interesting. Could the authors also analyze these mixtures with the 30rfu AT that was applied to the rest of the data to report how LRs change at the two ATs?

Section 2.4 – what are the race/ethnicity/genetic ancestry of the DNA donors and how was this information accounted for in selecting appropriate population allele frequencies for LR calculation?

Section 2.5 – the indicated population frequency databases from Budowle, Hill, and Moretti all utilize population defined by self-reported racial and ethnic backgrounds. How were samples from this study partitioned into the very resolved pop data from Moretti versus the broad pop definitions from Budowle?

Section 2.7 – please provide a reference supporting the use of different burnings for the number of persons per mixture. Is there an error measure associated with these burnin values? E.g., if 4 analysts evaluated the electropherogram and determined 2-person mixture but a 5th  determined a 3-person mixture, how many burnings should be applied?

Section 3.1 – it would be helpful for readers if the authors defined the variance factors and sigma measurements. A brief concluding sentence for this section would be useful. Why should the reader care about the variance and sigma data?

Section 3.1 – Supplementary Figures 1 & 2. Could the authors please plot these two images with the same y-axis scale? It is difficult to compare the performance of 30 and 50 rfus when the scales are different

Figure 1 –The font size of Figure 1 could be increased to improve clarity. Because the text makes comparisons between the undiluted and diluted mixtures (e.g., Figure 1e and 1f) it would be beneficial to show these data (i) on the same graph and (ii) using the same scale. The five-person mixture in Figures 1e and 1f cannot be compared graphically in the current form. Were any formal statistical tests performed to assess differences in mean LR given each condition?

Supplementary Table 1 – please reformat columns to use consistent notation of decimal position.

Supplementary Table 2 – which reference population was used to estimate allele frequencies for this calculation? What locus is being analyzed here?

Section 3.3 – it is unclear to me why only one sample was used to calculate accuracy of MaSTR using manual calculation. It would be more reliable to assess accuracy of MaSTR LRs for all 6 of the known DNA contributors. At least this was the authors can formally test for differences in mean per-locus RMP for each locus. Indicating observational similarity is not a very rigorous assessment of the data.

Section 3.3 – again it is unclear why only a subset of mixtures were run through MaSTR and EuroForMix. What is the rationale for not using all of the mixtures to get a better assessment of noise and differences between system? With only a few observations there is no way to reliably compare MaSTR to EuroForMix.

Section 3.3 – the authors may also consider that contributor 1683 is highly genetically admixed and does not fit the proposed population frequencies given to MaSTR.

Figure 2 – font size should be increased for better readability.

Figure 3 – font size should be increased for better readability. Figure 3d appears to have a confusing y-axis with micro symbol included. Please correct this. Figure 3a has a 1p and Figure 3c has a 100p that should be corrected to 10^x notation.

Figure 4 – font size should be increased for better readability.

Figures 5 and 6 – Should the figure legends indicate that the average peak height is for the minor contributor? If more than one contributor is included in each data point, how do the APH estimates account for this? Or have I misunderstood this graph?

Discussion and Conclusions – forgive me if I misunderstand how MaSTR handles reference data for allele frequencies but there is no discussion of how the allele frequency reference population is selected and how this selection might influence the resulting LRs. This should either be expanded upon or listed as a limitation of the validation study.

Reviewer 2 Report

MaSTR is a probabilistic genotyping software used for the interpretation of DNA mixtures and assign a likelihood ratio for a set of propositions.  To my understanding, MaSTR is a PG software that has been commercially available a few years ago.  But no developmental validation was published at the time.  This manuscript presents the internal/developmental validation of MaSTR to the forensic community, addressing the SWGDAM developmental validation guidelines - Sensitivity, Specificity, Precision,  and Accuracy.   While developmental validation of a PG software is not novel, the developmental validation of MaSTR is important to be considered for publication, not only for the software but also supports the community's uptake of PG software and continuous models.

Overall, the authors have presented a lot of analysis and in my opinion, have a well-structured paper.  However, at times, I feel it is a lot of quantity over quality in-depth analysis.  For example, Figures 5 through 7 is similar analysis, so some of this could sit in the supplementary to be more concise.  However, investigation of unexpected/outlying results could be more in-depth.  Such as the LR assigned to 1683 in EuroForMix and MaSTR.  I would have appreciated a more in-depth investigation into the diagnostics presented here.

Additionally, I think the plots need to be reconsidered to support the author's conclusions.  And possibly double-check the use of Type I & II errors.  (I personally find the use of false inclusion/exclusion easier language to understand).

Below are some additional comments for the authors to consider.

Page 1 - Fully continuous systems use allele data and additional information such as mixture ratio,  stutter percentages, and electropherogram peak heights.

I think mixture ratio is not needed here.  Most of the time, continuous systems will estimate the mixture ratio.  For example, in EuroForMix, it is one of the parameters that is estimated and in STRmix mixture ratio is estimated based on the template parameter for each contributor during MCMC.  Having said that, STRmix does have the ability to inform the interpretation with a prior expectation on the mixture ratio.  

Page 6 - Uniquely named mixture analysis Models were

Is Models the menu/module name? Clarification here.

Page 6 - 2.6. Protocol Data Set

Is the Protocol Dataset the same for the assessment of N-1 stutter peaks and the variance factor estimation?  Or are these two separate datasets?

Is there a variance factor for alleles?  

Page 6 - It should be noted that the MaSTR™ software calculates both the simple (no specified order of contributors) and overall (specified order of contributors) likelihood ratios; however, only the overall values [28]

I believe the community prefers the terms sub-source and sub-sub source LRs.  See Cook & Evett’s hierarchy of propositions (1998) and Evaluation of forensic genetics findings given activity level propositions: A review (2018).

Page 6 - Also, the version of MaSTR™ used in this study would not report an absolute exclusion (LR = 0), nor did it use a numerical cutoff limit for LRs <1.  MaSTR™ calculated the LR for each unknown contributor, whether entered as a reference or an alternate and reported that value no matter how low it was.

Why is this the case?  Consider a single locus unambiguous single-source evidence profile with the alleles 16, 18.  If the reference genotype at this locus is 20/20, would you not manually exclude (LR=0) this POI?

Page 8

See above, for sub-source vs. sub-sub source LRs.

Page 8 - At 50 RFUs the variance factor was ~4.6728, the linear variance factor (calculated using the linear degradation model) was ~3.7488, the exponential variance factor (calculated using the exponential degradation model) was 3.7661.

Given that the protocol data set (and the mixture dataset) does not have degraded samples maybe the degraded variance factors are not required here?  Otherwise, I question how reflective the degraded variance factors given your dataset doesn’t have degraded profiles.

Page 8 – Supp. Fig 1.

It is discussed that the variance factor is for backwards stutter.  Above, I queried if there is a variance factor for alleles.  In Supp. Fig 1 and 2 look like they are plotting the het-balance of allele.  I think clarification is required around the variance factor and how it applies to alleles and stutter. Or if they are they are given the same variance factor.

Page 8

There is also no discussion on the model results for back stutter.  E.g. the stutter ratios versus the allele designation, then determine the linear regressions.

Page 8-9

Box plots are inappropriate here.  Consider that there are five replicates for each analysis, for each box plot there will be five points: Min, 1Q, Average (or Median), 3Q, Max.  The LRs for the five replicates would effectively represent these five points on a Box plot.  I think scatter is more appropriate given the few data points.

Furthermore, for the higher-order mixtures, because the LRs are different for the different contributors, it distorts the Y-axis.  Therefore, this may give the illusion that the LRs are more precise than they are.

Page 10 - contributor 1683, which had one anomalous replicate test that yielded a value approximately three log units lower

While 3 log units lower may be acceptable in some cases.  Consider the difference between 10^3 and 10^6 on a verbal scale (moderate support vs. very strong support).  This is why I think a more in-depth explanation is required here.  Even if it is in the supplementary, I would have liked to see the authors try and isolate which locus may be causing the deviation in the LR, is there a genotype that MaSTR does not like in this analysis compared to the others?  

Page 11 - The “Quantitative LR (Bayesian based)” calculation in EuroForMix was selected for the analyses performed, as it corresponded most closely to the calculations used by MaSTR™.

What LR is reported by MaSTR? Sub-source or sub-sub source for this comparison?  It was noted above that LR calculations used NRC II 4.1.  I believe EuroForMix uses NRC II 4.2.  Unless the product rule is used here.  Some information about the EuroForMix setup is required here.  E.g. Is only back stutter modelled? Is degradation enabled?

Page 11 - An exception was con-tributor 1683 in the three-person 1:1:1 (~500:500:500 pg) mixture. MaSTR™ yielded an LR of 1.30E+04, while EuroForMix gave 7.23E+01.

On the verbal scale, this is a much larger difference, especially if the template is relatively high.  What were the mixture proportions proposed by both programs?  Was it close to the experimentally designed 1:1:1?  Or was it supporting Major:Major:Minor?  Were the diagnostic values in EuroForMix examined? Such as the PP-Plots?  Were there any diagnostic values to assess in MaSTR?  Were these unexpected?  EuroForMix uses 5/2N to deal with previously unobserved alleles in the allele frequency database, is this the same in MaSTR?

Page 11 – Neither software package demonstrated any Type I errors with the tested mixtures.

Type I errors are “False positives”.  In this section, the LRs were all assigned to known contributors to the mixture.  Therefore, I think perhaps the authors mean Type II error or “False negative” as none of the LRs assigned to the known contributor is less than 1.

If the software were tested against known non-donors to each mixture, and an LR > 1 was assigned, then that would be a Type I error?

Page 12 – Ratio Trace Plots

In the example above where the 1:1:1 three-person mixture gave divergent results between the two software, what did the ratio trace plot look like?  Was this to expectation?

Page 13 – Figure 3

I think the plots would be better if they followed something like Taylor (2014) for the non-contributors. (Using continuous DNA interpretation methods to revisit likelihood ratio behaviour).  This way we can see the expected trend of LRs for the template of the sample.  You can still show the different populations with different colors in plots such as those presented by Taylor (2014).  Also, I think the p, f, and mu make the plots more difficult to read.  I think keeping it as scientific numbers is sufficient for the audience of the journal article.

Page 13 - As the number of contributors and complexity of the undiluted mixtures increased, the LRs approached values near one, as expected.

This cannot be seen in Figure 3, if the authors replot like in Taylor (2014), then this conclusion may have some more merit.

Page 13 - However even the worst undiluted 

‘worst’ is subjective, maybe even the most complex undiluted mixture in our dataset - …

Page 14 -  Type II errors were seen in some of the random database results under extreme conditions of allele drop-out, as previously described (Figure 3).

Type I error?

Page 14 - The distributions of LRs for true contributors and non-contributors are shown in Figure 4.

Figure 4 shows the proportion of LRs for true contributors and non-contributors.  These LRs are binned into the verbal scale equivalents, so I believe they no longer show the distribution.  Above, the authors state that MaSTR does not have absolute exclusion nor have a cutoff.  By binning the LRs, the authors have effectively introduced this cutoff.

I think the plots in Figure 4 should be a density or frequency histogram.  That would give us a better idea of the “distribution”, but I suspect that this would be a multi-modal distribution as there are different samples for each group.  Personally, I think Figure 4 would be better represented by plots such as those in Taylor (2014), which the authors have done in part in Figure 5 onwards.

Page 19 - the quantity of contributor DNA template for amplification on the average calculated LRs for low template (≤100 pg) true con-tributors was examined for the three-, four-, and five-person mixtures. It should be noted that the major contributors in some of these mixtures were amplified with DNA template quantities >100 pg and thus data for them were not included, while contributors that made up the “middle” positions in mixture combinations often were, as they fell below the 100 pg cutoff.

in some of these

Why is this cutoff needed? Why not show all the results? I think it is valuable to show all the results.  If the authors are trying to show the low template end, I think the authors can consider taking the logarithm of the x-axis.

Page 19 - The average log(LR)s

Previously, the authors use the average LR, now the average log(LR) is introduced.  Please check that this is consistent because the average log(LR) would be the geometric mean.  Please reword where necessary.

Consider the 5 LRs 1000, 1100, 1200, 1300, 1400. The average LR is 1200. The log of this is 3.079.

However, the log of the 5 individual LRs is 3, 3.04, 3.08, 3.11, 3.15.  The average of the logged LRs is approx 3.076.

Figures 5-6

I think just keep one approach, either APH with or without obligate I think.  APH is an estimate of the template amount for that contributor.  I don't think both are required to draw similar conclusions about the trend.

I think Figure 7 could sit in the supplementary.  Also to me, it looks like Figure 7 has fewer data points in comparison because of overlap.  Some transparency (ggplot would be alpha=0.5) might help, or adding facets might help with the plot visualization.

Page 20 - The LRs for the three-person high allele sharing mixtures tended to be greater than those of most of the three-person low share mixtures. This was not surprising as allele stacking was more prevalent in the high share mixture combinations and thus more alleles had greater peak heights and fewer instances of drop-out, even though the mixtures had the same number of contributors as well as the same amounts of template DNA.

I don't think there is enough data here to support this conclusion/statement.

The NoC section. I think instead of a table, it would be better to present this as a plot where we are looking at the N+/-1 logLRs versus the N logLRs.  The authors could use the color/shapes to indicate the type of mixture.  I think the authors should also include the results from non-donor LRs to demonstrate N+1 or N-1's effect on the non-donor LRs.

Page 28 - The MaSTR™ software demonstrated a very high degree of sensitivity and specificity. [...] These results show that MaSTR™ performs similarly to other probabilistic software regarding Type II errors [21].

"very high degree" is subjective. Consider re-wording. Perhaps to better demonstrate the degree of sensitivity and specificity or the discriminatory power, consider showing a ROC plot.  Also because the validation datasets are different, and different kits/parameters etc, I think it would be difficult to say that MaSTR performs similarly to other PG software, as it's not a direct comparison.  For example [21] uses Fusion 6C which has a couple more loci than 5C.

Page 28 - much larger number of in silico true non-contributor genotypes resulted in LRs>1. These Type II errors occurred in mixtures of three or more persons with low DNA template con

Type I (False positive?) error?

Reviewer 3 Report

The study tested MaSTR software performance for evaluating 2- to 5-person mixtures at different conditions. It is a valuable study because forensic community needs this kind of softwares for calculating LR in complex mixtures, but in order to be used in caseworks, they need to be carefully validated.

Authors did an extensive work to evaluate the MaSTR software performance. Only a few comments about some topics.

First one: the Bayesian based quantitative LR estimation in Euroformix, the one the Authors used fo the concordance study,  is not very robust and the developpers theirselves suggest to use the MLE based one, the one that is validated. Did the Authors try to perform the calculation with this method and check if results are concordant? 

Second one: the Authors performs the evaluation of the replicates indipendently. If the software can handle replicates all togheter, why did the Author choose not to perform the evaluation of the replicates all together? It has been demonstrated that considering replicates together can reduce Type I and II errors.

One more comment about considering POI's relatives. Why did the Authors choose not to perform this evaluation in the internal validation study? Is it time-consuming? It would have add value to the study because relatives could create problems in the interpretation of complex mixtures.

Round 2

Reviewer 1 Report

In my opinion, the authors have not made any serious effort to improve the quality of their work and have instead made every effort to avoid editing their manuscript in response to feedback. Relative to my first round of critiques, I have several follow up points that must be clarified prior to publication of this work in Genes or elsewhere.

In Figure 1, why are 40k and 80k MCMC iterations mentioned in the results of Figure 1a but there are no labels for 40k and 80k in the figure itself? Should the x-axis labels be 40k and 80k instead of 5k and 10k? The same is true for mention of 400k iterations and 160k iterations – where are these data in Figure 1? These need to be more clearly labeled. As a reviewer looking for sound science in the submission, author feelings about a result are not a viable metric upon which to write or publish a paper. Admittedly, Figure 1 not only has poor image quality, but the comparisons made for each sample’s LR are so minor that they verge on meaningless unless the authors can show a statistical difference between means/variances/etc. Figure 1c and 1d for example show almost the same distribution for each sample at each iteration…which iteration is better? How do you know? 

Regarding accuracy of MaSTR: which published studies are the authors referring to? I don’t think this being an internal validation changes the fact that if the authors want to convey accuracy of the system, there should be a comprehensive assessment of accuracy. The approach applied by the authors reads as lazy and inappropriate unless there are references showing a near 1-to-1 relationship between MaSTR LRs and by-hand LRs.

Regarding MaSTR vs. EuroForMix: again this is a lazy approach to an important question. Without showing that all data were run on both platforms, one could reasonably assume that the authors did run all of the data and selected only the “good” results to support their claims. I as a reviewer have no way of knowing if this is or is not the case without seeing the results.

Regarding all screen capture images: surely raw data can be downloaded from the software to recreate high quality images that are legible and clear. This is a poor justification for the submission of low-quality images that cannot be read or interpreted by the target audience. 

Author Response

In Figure 1, why are 40k and 80k MCMC iterations mentioned in the results of Figure 1a but there are no labels for 40k and 80k in the figure itself? Should the x-axis labels be 40k and 80k instead of 5k and 10k? The same is true for mention of 400k iterations and 160k iterations – where are these data in Figure 1? These need to be more clearly labeled. As a reviewer looking for sound science in the submission, author feelings about a result are not a viable metric upon which to write or publish a paper. Admittedly, Figure 1 not only has poor image quality, but the comparisons made for each sample’s LR are so minor that they verge on meaningless unless the authors can show a statistical difference between means/variances/etc. Figure 1c and 1d for example show almost the same distribution for each sample at each iteration…which iteration is better? How do you know? 

The distinction between total number of MCMC iterations and iterations per chain was not entirely clear and the text in the figure caption and in the body of the paper has been changed to better delineate the two.

Levene’s tests for variance were performed and no significant differences were found with this data.  Text has been added describing this and changes were made to the section to reflect the information.

Regarding accuracy of MaSTR: which published studies are the authors referring to? I don’t think this being an internal validation changes the fact that if the authors want to convey accuracy of the system, there should be a comprehensive assessment of accuracy. The approach applied by the authors reads as lazy and inappropriate unless there are references showing a near 1-to-1 relationship between MaSTR LRs and by-hand LRs.

The section on accuracy has had additional information added.  The RMPs of all of the genotypes used to create mixtures were calculated by hand and by MaSTR for concordance comparison and that data was added in text and a supplementary table.  All of the loci and the complete profile LRs for the major contributor to the two-person mixture described in the accuracy section were hand calculated with and without the weighted probabilities and compared for concordance with the same data from MaSTR and text, as well as a supplementary table was added for that data.  Modeling performance for MaSTR was examined using LR trends for different mixture ratios and DNA template quantities for the two-person mixture to examine if MaSTR followed the expected changes for the major and minor contributors’ LR values.  Text and supplementary figures were added for this information.

Regarding MaSTR vs. EuroForMix: again this is a lazy approach to an important question. Without showing that all data were run on both platforms, one could reasonably assume that the authors did run all of the data and selected only the “good” results to support their claims. I as a reviewer have no way of knowing if this is or is not the case without seeing the results.

The comparison between MaSTR and EuroForMix is intended to supplement the other accuracy assessments, especially for 3 and 4 person mixtures that cannot be effectively done by hand.  Running all of the study’s data on both systems is well outside the scope of the study and is not a normal feature of other validation studies of this type.  We are showing all of the data we have that was analyzed with both systems and point out differences in results in the text, as well as show the data in table form so all readers can see the specific values.  We did not pick or choose “good” or “bad” data, there is just the data we have, which is fully intended to be limited.  We have however, amended the text to indicate that this was designed as limited comparison.

Regarding all screen capture images: surely raw data can be downloaded from the software to recreate high quality images that are legible and clear. This is a poor justification for the submission of low-quality images that cannot be read or interpreted by the target audience. 

After checking with the software developer, we confirmed that the raw data for the synthetic mixtures and mixture proportions are not stored by the software other than in the plots generated.  In order to address the concerns regarding the readability of the images, they have been re-sized and re-imaged for better clarity.

Reviewer 2 Report

Page 6 - length of burn-In and Thin N values, as well as number

Capitalization

Page 8 - Variance factor values for allele peaks,, including backwards stutter peaks,

Comma

Figure 1

The dotted horizontal line is not explained

Author Response

Page 6 - length of burn-In and Thin N values, as well as number

Capitalization

Capitalization has been corrected. 

Page 8 - Variance factor values for allele peaks,, including backwards stutter peaks,

Comma

Punctuation corrected. 

Figure 1

The dotted horizontal line is not explained

Apologies for this oversight. This line indicates the overall mean of all of the data in the plot. Text has been added to explain this.